# Sensitivity analysis of an aerosol aware microphysics scheme in WRF during case studies of fog in Namibia

Michael Weston[1], Stuart Piketh[2], Frédéric Burnet[3], Stephen Broccardo[2*], Cyrielle Denjean[3], Thierry Bourrianne[3] and Paola Formenti[4]

[1] Research and Development Division, Khalifa University, Abu Dhabi, United Arab Emirates
[2] School of Geo- and Spatial Science, North-West University, Potchefstroom, South Africa
[3] CNRM, Université de Toulouse, Météo-France, CNRS, Toulouse, France
[4] Université Paris Cité and Univ Paris Est Creteil, CNRS, LISA, F-75013 Paris, France
[*] Now at: Bay Area Environmental Research Institute / NASA Ames Research Center, CA, United States

*Correspondence to*: Stuart Piketh (stuart.piketh@nwu.ac.za)

**Abstract:** Aerosol aware microphysics parameterisation schemes are increasingly being introduced into numerical weather prediction models, allowing for regional and case specific parameterisation of cloud condensation nuclei (CCN) and cloud droplet interactions. In this paper, the Thompson aerosol aware microphysics scheme, within the Weather, Research and Forecasting (WRF) model, is used for two fog cases during September 2017 over Namibia. Measurements of CCN and fog microphysics were undertaken during the Aerosol, Radiation and Clouds in southern Africa (AEROCLO-sA) field campaign at Henties Bay on the coast of Namibia during September 2017. A key concept of the microphysics scheme is the conversion of water friendly aerosols to cloud droplets (hereafter referred to as CCN activation), which could be estimated from the observations. A fog monitor 100 (FM100) provided cloud droplet size distribution, number concentration ($N_t$), liquid water content (LWC) and mean volumetric diameter (MVD). These measurements are used to evaluate and parameterise WRF model simulations of $N_t$, LWC and MVD. A sensitivity analysis was conducted through variations to the initial CCN concentration, CCN radius and the minimum updraft speed, important factors that influence droplet activation in the microphysics scheme of the model. The first model scenario made use of the default settings with a constant initial CCN number concentration of 300 cm$^{-3}$ and underestimated the cloud droplet number concentration while the LWC was in good agreement with the observations. This resulted in droplet size being larger than the observations. Another scenario used modelled data as CCN initial conditions which were an order of magnitude higher than other scenarios. However, these provided the most realistic values of $N_t$, LWC, MVD and droplet size distribution. From this it was concluded that CCN activation of around 10 % in the simulations is too low, while the observed appears to be higher reaching between with a mean (median) of 0.55 (0.56) during fog events. To achieve this level of activation in the model, the minimum updraft speed for CCN activation was increased from 0.01 to 0.1 ms$^{-1}$. This scenario provided $N_t$, LWC, MVD and droplet size distribution in the range of the observations with the added benefit of a realistic initial CCN concentration. These results demonstrate the benefits of a dynamic aerosol aware scheme when parameterised with observations.

**Keywords:** fog; Namibia; WRF; microphysics; aerosol aware; Thompson scheme; CCN; FM100

## 1 Introduction

The Central Namib Desert is situated along a narrow coastal area about 100 km wide on the Namibian Coast. It is adjacent to the cold Benguela current in the South Atlantic Ocean and consequently experiences fog when the moist air from above the ocean is advected over the desert surface. As a vital source of fresh water in this arid ecosystem,

fog has been a topic of study for decades (Cermak, 2012; Lancaster et al., 1984; Olivier, 1995; Seely and Henschel, 1998; Seely and Hamilton, 1976), and has recently motivated long-term distributed observations (the FogNet network which is predominantly meteorological and radiation measurements; (Muche et al., 2018)) as well as intensive dedicated field campaigns (Spirig et al., 2019) that have shed new light on the spatial and temporal characteristics and dynamics of fog in the region (e.g. (Andersen et al., 2019)). Fog occurs predominantly at night and early morning hours and is most frequent closer to the coast, reaching about 120 days a year and decreasing to 40 (5) fog days at sites 40 (100) km inland (Olivier, 1995; Spirig et al., 2019; Andersen and Cermak, 2018; Cermak, 2012). Fog is most frequent in winter (May - August) at coastal sites and in summer (September - December) at inland sites (Andersen et al., 2019; Lancaster et al., 1984; Spirig et al., 2019; Olivier, 1995; Nagel, 1962). The fog is predominantly advective and high in elevation corresponding to low stratus clouds that intersect with the land (Andersen et al., 2020; Andersen et al., 2019). This is possible in the central Namib as the elevation gradually increases from the coast to 1000 meters above sea level (m.a.s.l) at the escarpment about 100 km away to the east. In summer months the stratus layer occurs at around 500 m.a.s.l (Andersen et al., 2019) and allows the fog to penetrate further inland. In winter, the stratus layer is lower, which limits fog to the coastal area.

Fog formation and lifetime is dependent on the atmospheric thermodynamic (radiation, turbulence and mixing) and surface conditions (albedo, soil characteristics, roughness length, moisture content). Therefore, the simulation of fog first requires that state atmospheric variables related to temperature and moisture are simulated adequately by models, which implies representing the coupling of land-atmosphere interactions and good parameterisation schemes of the planetary boundary layer (PBL) (Bergot and Lestringant, 2019; Boutle et al., 2018; Juliano et al., 2019; Maronga and Bosveld, 2017; Steeneveld and De Bode, 2018).

Additionally, forecast of fog requires improved parameterisation of its microphysical properties (e.g. (Bott, 1991; Gultepe and Milbrandt, 2007; Tardif, 2007)). Several studies have demonstrated the importance of the cloud condensation nuclei (CCN) and cloud droplet relationship in simulating the fog life cycle (e.g. (Boutle et al., 2018; Maalick et al., 2016; Stolaki et al., 2015)). The formation of fog depends on the capabilities of the pre-existing aerosols to act as CCN thereby providing a substrate for water vapour to condense and grow to form a fog droplet. The properties of the CCN aerosols play a role in shaping the microphysics of the fog droplets (Gultepe and Milbrandt, 2007; Haeffelin et al., 2013). The size distribution of the droplets has an important effect of the radiation balance (Boutle et al., 2018; Egli et al., 2015; Mazoyer et al., 2019; Poku et al., 2019). Mazoyer et al. (2019) showed a widening of the droplet size distribution (DSD) towards the fog top as droplets grew by collision and coalescence. Egli et al (2015) showed that liquid water content (LWC) varies through the fog layer, usually with a maximum near the centre of the fog layer. In this case, the increase in LWC was due to an increase in number concentration and not the conversion of small droplets to larger droplets. These complex microphysical processes within the fog layer are increasingly being introduced into model simulations (Thompson and Eidhammer, 2014; Wilkinson et al., 2013).

Aerosol-aware microphysics schemes, with the main function of representing grid scale clouds in a simulation (Thompson and Eidhammer, 2014; Wilkinson et al., 2013), are now being used to represent fog in the lowest model level and are showing encouraging results. Boutle et al. (2018) and Poku et al. (2019) demonstrated improved fog simulations of cloud droplet number and sedimentation based on model sensitivity to CCN concentrations in an aerosol aware scheme. Mazoyer et al. (2019) demonstrated that improving supersaturation and activation of CCN ameliorated forecast results of droplet concentration. As these schemes advance, more detailed information on CCN size distribution, chemistry and activation can be provided as input to models, in particular in numerical weather prediction

(NWP) models, when they are often represented by implementing a lookup table of CCN activation (see (Ghan et al., 2011; Saleeby and Cotton, 2004).

Microphysics schemes in mesoscale models are designed with cloud formation in mind, and not necessarily fog formation. Droplet activation is based on a parcel that is lifted and cooled adiabatically to reach saturation, and is therefore most sensitive to the updraft speed. However, fog often occurs under stable conditions where updrafts are low in speed or even negative. Thus, droplet activation is dependent on other processes like non-adiabatic cooling. This point is highlighted by Boutle et al. (2018) who observe a cooling rate prior to fog formation of 1 Khr$^{-1}$ which is

equivalent to an updraft speed of 0.04 ms$^{-1}$ assuming a wet adiabatic lapse rate of 6.5Kkm$^{-1}$. In their case, the minimum updraft speed in the microphysics scheme was 0.1ms$^{-1}$, and thus would over estimate fog drop activation. To address this issue, Poku et al. (2021) expanded an existing microphysics scheme to allow for non-adiabatic cooling, which allowed for more realistic cloud droplet number concentration in the simulation.

In this paper, we assess an aerosol aware microphysics scheme that uses a minimum updraft speed of 0.01 ms$^{-1}$, which

equates to a cooling rate of 0.23 Khr$^{-1}$. This minimum updraft speed should solve some of the over activation issues highlighted by Boutle et al. (2018) and Poku et al. (2019). The main aim is to see how this scheme performs, before applying major changes in the code to account for non-adiabatic cooling rates or similar. Furthermore, our study site is located in the tropics which we see this as a benefit to the community at large as most fog modelling studies are focused on mid- to high-latitude sites.

In this paper we present an assessment of the capabilities of the Weather Research and Forecasting (WRF) model to predict two fog events observed in September 2017 along the coast of Namibia. WRF has been used to simulate fog previously using rule based methods (Román-Cascón et al., 2016; Weston et al., 2021), however, we will focus on the aerosol aware microphysics scheme capabilities in the model. The area is interesting for studying fog formation as the contribution of anthropogenic sources to the background aerosol concentration is limited, meaning that pollution is

minimal. Namibia has a population density of 3 people km$^{-2}$ (Statista, 2020), and previous research has shown that the influence of anthropogenic activities is minor (Formenti et al., 2019; Klopper et al., 2020).

Fog is diagnosed from the model using the LWC from an aerosol aware microphysics parameterisation scheme. This work takes advantage of the measurements of surface level fog microphysics that were performed at a ground-based site on the coast of Namibia as part of the Aerosol, Radiation and Clouds in southern Africa (AEROCLO-sA)

campaign (Formenti et al., 2019).

The work is structured as follows. In section 2 a description of the study sites, data sets and model configuration is presented. Section 3 includes the model results and discussion. A summary of the main findings is given Section 4.

## 2 Methodology

### 2.1 Study area

The Central Namib is a coastal desert on the coast of Namibia. It lies between the Atlantic Ocean to the West and the escarpment over 1000 m in elevation about 100 km to the east. The mean annual rainfall is highest at the escarpment (100 mm) and decreases towards the coast (< 50 mm) (Lancaster et al., 1984; Spirig et al., 2019). The land cover type undergoes a stark transition at the ephemeral Kuiseb River with the rocky Gravel Plains to the north and large sand dunes to the south. Surface air flow along the coast is dominated by an onshore wind, predominantly from the south-

west (Lindesay and Tyson, 1990). This flow can be amplified by the synoptic scale circulation of the South Atlantic

High pressure system. However the onshore flow does not penetrate far inland where the surface air flow is controlled by the mountain-plain wind. The easterly wind is much drier and limits moisture transport inland. As a result, fog forms in a narrow band along the coastline.

**2.2 Model Configuration**

The Weather Research and Forecast model (WRF v3.9.1, Skamarock et al, 2008) was used to forecast next day fog. Three nested domains with horizontal grid resolutions of 27-9-3 km were defined and the model was run with one-way nesting (Fig. 1). The parent domain, domain 1, was sufficiently large to allow the movement of low pressure cells in the easterlies and westerlies to pass through and provide boundary conditions to the nested domains. Domain 2 extended 4266 km from east to west and 2296 km from north to south. Domain 3 was constructed with the study site

approximately in the centre and extended 1386 km from east to west and 720 km north to south. A total of 50 vertical levels were used with extra vertical levels added near the surface to allow for 11 model levels below 500 m above ground level (a.g.l). This was decided after initial simulations demonstrated that the default vertical resolution was to coarse near the fog top. The mean height of the lowest 5 levels was 34, 71, 109, 146 and 184 m a.g.l. Boutle et al. (2022) evaluated results from large eddy simulation (LES) and single column models (SCM) for a radiation fog case

in the United Kingdom and recommend having a first vertical level less than 10 m and six or more levels below 150 m. An increase in vertical resolution is expected to better simulate strong moisture and temperature gradients in the lower troposphere (e.g. (Branch et al., 2020)). However, Ajjaji et al. (2008) reported that an increase in vertical resolution can have the opposite effect and inhibit cloud formation for fog events over the United Arab Emirates, an arid region similar to Namibia, during a WRF real case (i.e. not SCM) simulation. Furthermore, our set up is in line with the

vertical profiles reported in the literature which show that the moisture is trapped below 500 m (e.g. (Andersen et al., 2019; Formenti et al., 2019; Spirig et al., 2019)). The model was initialised at 06 UTC (08h00 local) with Global Forecast System (GFS v14) data at 0.25 degree resolution and updated every 6 hours (Ncep, 2015).

Sea surface temperature (SST) from the GFS was allowed to follow a diurnal cycle in WRF using the method described by Zeng and Beljaars (2005). This means that 6 hourly values from the GFS are interpolated to provide hourly updated

values in WRF. The Noah land surface model was used with land cover classes from the United States Geological Survey (USGS) (Loveland et al., 2000; Sertel et al., 2010). The default soil texture in WRF is from the State Soil Geographic (STATSGO)/Food and Agriculture Organization (FAO) soil database (Dy and Fung, 2016; Sanchez et al., 2009).

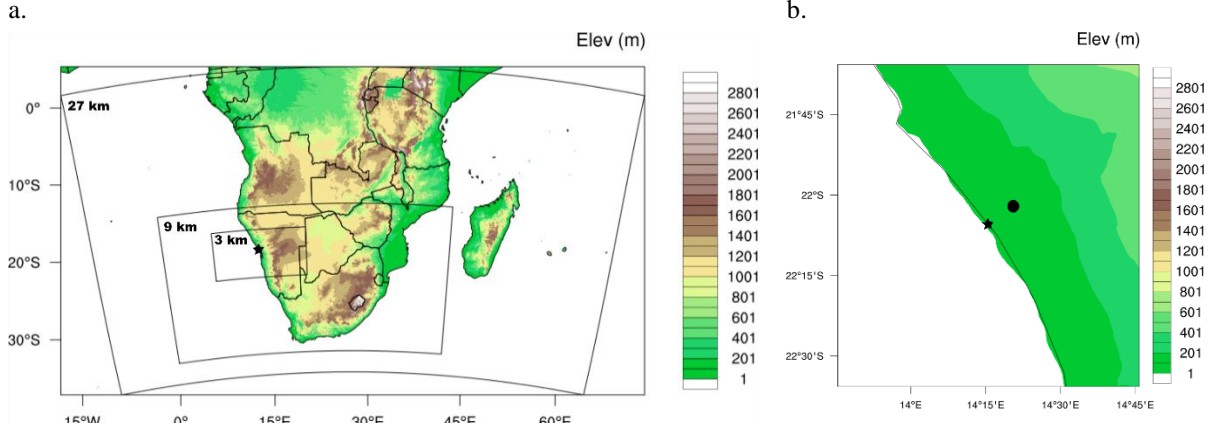

**Figure 1: a.) WRF model domains and b.) insert showing the Henties Bay site (star) and inland point (circle) from the model which is used during the evaluation of the simulations.**

Shortwave and longwave radiation was controlled by the rapid radiation model for general circulation model applications (RRTMG, (Iacono et al., 2008)). The revised MM5 scheme was used for the surface layer and the Mellor-Yamada Nakanish and Niino Level 2.5 (MYNN2.5) scheme for the planetary boundary layer (PBL) physics parameterisation (Nakanishi and Niino, 2006). The Kain-Fritsch Cumulus physics was activated in domain 1 but deactivated in domain 2 and 3 (Kain, 2004). This allows for sub-grid scale precipitation in the coarse resolution parent domain allowing for realistic water balance when applying the one-way nesting.

### 2.2.1 Fog microphysics

The grid-scale cloud microphysics bulk scheme by Thompson and Eidhammer (2014) assumes a gamma distribution of cloud DSD for droplet diameters, *D*, comprised between 1 to 100 μm. It is worth noting that the scheme accounts for conversion of cloud droplets to rain droplets, which are treated separately and the rain DSD spans a wider range in diameter. The number concentration per cloud droplet diameter, N(D) is calculated as:

$$N(D) = \frac{N_t}{\Gamma(\mu + 1)} \lambda^{\mu+1} D^\mu e^{-\lambda D} \tag{1}$$

where $N_t$ is the total cloud droplet number concentration (cm$^{-3}$), $\Gamma$ is the gamma function, $\mu$ is the shape parameter and $\lambda$ is the slope. It is a double moment scheme, meaning that cloud droplet mass and number concentration ($N_t$) is calculated from the DSD. The shape and slope parameters can be derived from $N_t$ and liquid water content (LWC) as follows:

$$\mu = min\left(\frac{1000}{N_t} + 2,15\right) \tag{2}$$

Where $N_t$ is in cubic centimetre (cm$^{-3}$).

$$\lambda = \left[\frac{\pi}{6} \rho_w \frac{\Gamma(4 + \mu)}{\Gamma(1 + \mu)} \left(\frac{N_t}{LWC}\right)\right]^{\frac{1}{3}} \tag{3}$$

Where $\rho_w = 1000$ kg m$^{-3}$ is water density and LWC is in kg cm-3.

The median volume diameter (MVD) is calculated as:

$$MVD = \frac{3.672 + \mu}{\lambda} \tag{4}$$

Fog was diagnosed when liquid water content was present in the lowest model level. This is a reasonable assumption for simulations with a suitably high resolution (i.e. observed fog feature is larger than model grid cell) and a sophisticated microphysics scheme (Zhou et al., 2012).

**2.2.2 Configuration of the aerosol aware sensitivity study**

The Thompson aerosol aware microphysics scheme (Thompson & Eidhammer, 2014; hereafter T14) was activated in all three domains to represent grid scale cloud microphysics. This novel scheme accounts for the initial number concentration, mean radius and hygroscopicity (kappa) of CCN, CCN activation to cloud (and other hydrometeor) droplets, which ultimately determine the maximum number of cloud droplets. Although the default settings for these variables are based on observations and literature, users can refine the values based on their study area. For example,

the initial CCN number concentration is set to 300 $cm^{-3}$ at the lowest model level. This may not be appropriate for a particular study region or event and can be adjusted accordingly. Furthermore, the option exists to replace initial CCN with user generated 3-dimensional data fields of CCN as input. An example is provided with the model based on a 7-year climatology of aerosols produced by the NASA GEOS-4 model (Colarco et al., 2010). A key approach of this scheme is the conversion of hygroscopic CCN, sometimes referred to as water friendly aerosols, to cloud droplets.

This is what is referred to as CCN activation in this document, which can be thought of as the ratio of cloud droplets to hygroscopic CCN. CCN activation is based on parcel method simulations by Eidhammer et al. (2009). These simulations were used to create a look up table (available within WRF) that vary the CCN activation to cloud droplets as a function of CCN number concentration, updraft speed, ambient temperature, CCN mean radius diameter and kappa (hygroscopicity) values (Fig. 2). It is important to keep in mind that CCN activation in the scheme assigns a

minimum updraft speed of 1 $cm \, s^{-1}$ when air is saturated. This allows for droplet activation under saturated conditions via the look up table even when the simulated updraft is negative. An additional novelty of this scheme is that it allows feedback of CCN and cloud droplets to the radiation schemes in shortwave and longwave, which was activated in the model simulations.

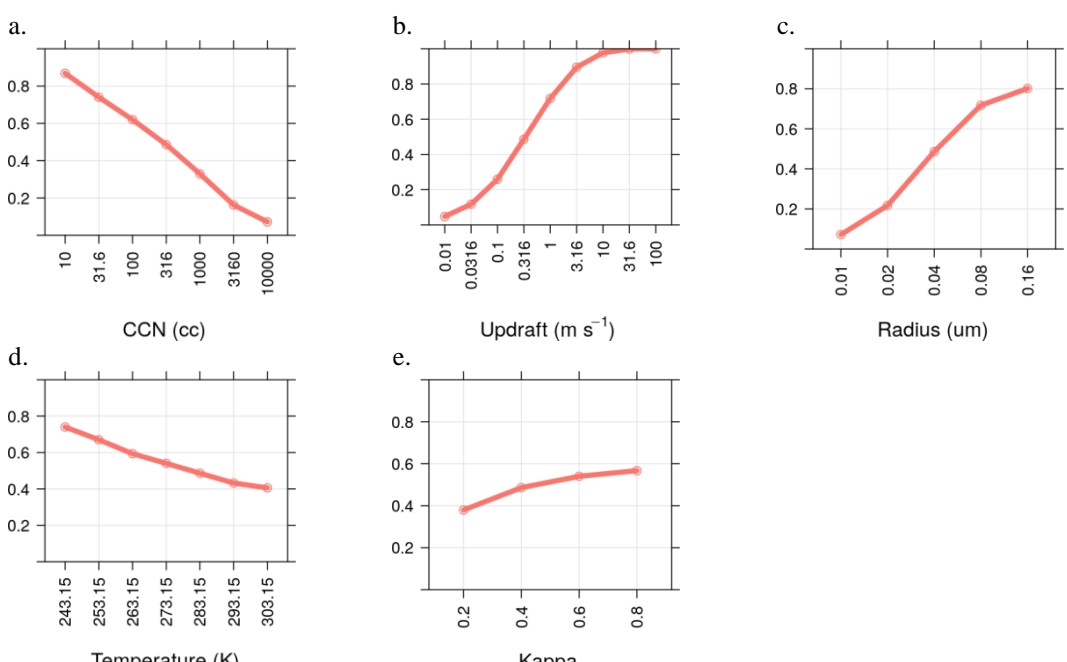

**Figure 2: CCN activation look up table. Activated fraction in response to a.) CCN concentration b.) updraft speed, c.) CCN radius, d.) ambient temperature and e.) kappa (hygroscopicity). For each plot, the remaining**

**four variables are kept constant as follows: CCN=316 cm$^{-3}$, updraft=0.316 ms$^{-1}$, radius=0.04 µm, temperature=283.15 K and kappa=0.4.**

Model scenarios were based on configuration of the microphysics, where the default CCN radius is 0.04 µm and the kappa is 0.4 (Fig. 3). Scenario 1 used the default initial CCN values of 300 cm$^{-3}$ near the surface and 50 cm$^{-3}$ in the free troposphere (hereafter CCN_300). The scheme applies a vertical profile to CCN concentration allowing concentrations to decrease exponentially from the surface to the free troposphere (Wrf Users Page, 2020; Fonseca et al., 2021)). In summary, the scheme assigns highest CCN concentration near the surface and follows an exponential decrease through the boundary layer to the minimum bound (50 cm$^{-3}$ in our case) which is then assigned to the lower free troposphere. The depth of the boundary layer is made to vary for different terrain heights, ranging to about 1000 m at sea-level to less than 100 m where terrain is greater than 2500 m. The thinner boundary layer at increased terrain height would have a steeper drop off in CCN concentration with height and subsequent dilution during day time mixing. In CCN_300 the initial CCN concentration over land and ocean is the same. Thompson and Eidhammer (2014) proposed different initial CCN concentration for land (300 cm$^{-3}$) and ocean (100 cm$^{-3}$) based on observations, as ocean air is generally cleaner and contains fewer CCN than continental air (Seinfeld and Pandis, 2016). We implemented this proposal for scenario 2 (hereafter CCN_300_landsea). The same treatment is applied to the vertical profile of CCN concentration as CCN_300. Scenario 3 is the default model setting using the 3-D climatology CCN modified from Colarco et al. (2010) (hereafter CCN_C10). As this is a 3-D data set, no idealised vertical profile is applied at initialisation. Simulations were run from 7 to 10 September 2017 to coincide with the best data from the FM100 instrument. Recall, simulations are initialised at 06 UTC and were run for 48 hours. The first 6 hours are discarded as model spin up (i.e. day 0 06 – 11 UTC). The next 24 hours in the model are used to assess the next day fog (i.e. day 0 12 to day 1 11 UTC).

A sensitivity study to CCN activation was carried out through permutations to the CCN and minimum updraft speed. Initial results from the study site showed that CCN could reach up to 500 cm$^{-3}$ near the surface (Formenti et al., 2019). Thus, the initial CCN number concentration near the surface was increase from 300 cm$^{-3}$ to of 500 cm$^{-3}$ (hereafter CCN_500), while the free troposphere was kept at 50 cm$^{-3}$. CCN_500 has the same vertical profile treatment as CCN_300. Initial concentrations over land and the ocean were the same, as in CCN_300. In another scenario, the mean radius of the CCN was decreased from 0.04 to 0.02 µm as part of a sensitivity analysis (CCN_300_r0.02 hereafter). All other setting were identical to CCN_300. In the final scenario the minimum updraft speed was increased from 0.01 m s$^{-1}$ to 0.1 m s$^{-1}$ (CCN_300_w0.1). This motivation for this scenario was to push the model to a higher CCN activation and see how this effects the size distribution results. For the last scenario, the minimum updraft speed was only assigned in the three lowest vertical levels in the model, and should not cause erroneous cloud at higher model levels.

Comparative statistics of model and observed cloud droplet count, LWC and MVD are presented in the results section. The model grid point that is closest to the site location is extracted for comparison with the observations. A second model grid point, hereafter referred to as "inland", is extracted about 13 km inland when travelling perpendicularly to the coast to demonstrate the dynamics and gradients in the model (Fig. 1b).

**Table 1: WRF model scenarios summary. Updraft refers to minimum updraft speed.**

|   | Scenario | CCN (cm$^{-3}$) | Kappa | Radius (µm) | Updraft (m s$^{-1}$) |
|---|----------|-----------------|-------|-------------|----------------------|
| 1 | CCN_300 | 300 | 0.4 | 0.04 | 0.01 |
| 2 | CCN_300_landsea | 300/100 | 0.4 | 0.04 | 0.01 |
| 3 | CCN_C10 | Colarco 2010 | 0.4 | 0.04 | 0.01 |

| 4 | CCN_500 | 500 | 0.4 | 0.04 | 0.01 |
| 5 | CCN_300_r0.02 | 300 | 0.4 | 0.02 | 0.01 |
| 6 | CCN_300 _w0.1 | 300 | 0.4 | 0.04 | 0.1 |

## 2.3 Observations

### 2.3.1 Meteorology and Microphysics

The AEROCLO-sA ground-based field campaign was conducted from 23 August to 12 September 2017 at the University of Namibia Campus in Henties Bay (-22.09495 °S, 14.2591 °E; (Formenti et al., 2019)). The campus is located on the coast line (elevation 20 m.a.s.l) and at the mouth of the non-perennial Omaruru River.

Measurements pertinent to this study include the meteorological measurements of temperature (2 m), relative humidity (2 m), wind speed (10 m), wind direction (10 m) from a Cimel Electronique Compact Weather Station part of the PortablE Gas and Aerosol Sampling Units (PEGASUS) mobile platform (Formenti, 2020b). Meteorological data was supplemented with atmospheric pressure from the control WRF model simulation described later on. This allowed further variables to be derived like water vapour mixing ratio and air density. Radiosonde measured atmospheric profiles of pressure, temperature, relative humidity, wind speed and wind direction (Formenti, 2020a). Radiosonde were launched form two locations, one being at the study site and the other being at Jakkalsputz, about 12 km south along the coast of the study site. Visibility measurements were available from a transmissometer with data recorded every 5 seconds. The transmissometer is a bistatic system, set up with the emitter and receiver 8.0m apart and approximately a metre off the ground. The instrument's analogue voltage output response to optical depth (OD) was calibrated using a series of stacked neutral density filters, and found to be linear. This voltage was recorded on a Campbell Scientific CR1000 data-logger. From the OD, we calculate the extinction coefficient and hence visibility using Koschmeider's law. Although this is known to have drawbacks (Lee and Shang, 2016; Nebuloni, 2005), we use the calculated visibility in a qualitative sense in this study

Cloud droplet measurements were conducted with a Fog Monitor 100 (FM100) providing droplet number concentration, size distribution and liquid water content for particle sizes with optical equivalent diameter of 1 to 50 μm. This instrument is a forward scattering spectrometer probe, where the droplet size is calculated based on scattered light from a laser and employing Mie theory (Spiegel et al., 2012). The particle is assumed to be spherical and made of water with a known refractive index. Droplets in the size range 2 - 50 μm can be counted with bin sizes between 2 and 3 μm. The instrument was calibrated with glass beads of known diameter and refractive index. The LWC is calculated based on the assumption that each droplet is spherical. Data was recorded every second and later aggregated to 1-minute averages.

A cloud condensation nucleus (CCN) counter (mini-CCNC) was deployed at the site. The super saturation during data capture was set to scanning mode, meaning that super saturation varied between 0.1 and 0.7 %. The CCNC was calibrated prior to the campaign using ammonium sulfate to determine the relationship between the temperature gradient along the column and the effective supersaturation. A wide, wind oriented intake facilitated air flow into the instrument and total CCN was recorded at a high frequency (every second).

### 2.3.2 Activated CCN

Activated CCN, the percentage of hygroscopic CCN that are present as cloud droplets as described in the model microphysics section, can be estimated as measurements of cloud droplets (from FM100) and CCN (from mini-CCNC) are co-located. This definition is maintained in the processing of the observations to allow for consistency in

comparing the model microphysics scheme with observations. Data overlap from these instruments coincided with a fog event on 9 September 2017, which was used to estimate the activated CCN at the site. The CCN counter cycled through super saturation from 0.1 to 0.7 % to account for varying aerosol sizes. For our purposes, CCN data was subset to super saturation range from 0.098 to 0.151 %. Practically, this means that CCN data is available at about 3-minute intervals due to super saturation cycle of the instrument. The 1-second data was then averaged to 1-minute for both the CCN and FM100 and paired according to matching times. CCN concentration is expected to be underestimated during wet conditions due to the design of the inlet on the instrument. To overcome this, we use the CCN concentration in the period prior to fog in the activated CCN calculation. A period of 1 hour of observations when visibility was 10 km prior to fog formation was used as the CCN sample. This period occurred from 01h00 to 02h05 UTC on 9 September 2017. The average CCN concentration was calculated for this period and used in conjunction with the Nt during fog conditions to calculate activated CCN as Nt/CCN (#cm$^{-3}$). Lastly, the data were filtered for conditions where Vis <= 1 km (based on the World Meteorological Organization definition; (World Meteorological Organization, 2008)) and $N_t$ > 25 cm$^{-3}$ where the visibility threshold meant that fog conditions were represented, and the $N_t$ threshold was estimated as being representative of a baseline in the fog monitor.

**2.3.3 Satellite**

The spatial evolution of the cloud and fog is presented using the Spinning-Enhanced Visible and Infrared Imager (SEVIRI) from Meteosat Second Generation 3 (MSG3) (Schmetz et al., 2002). MSG3, also known as the Meteosat 10 satellite, in geostationary orbit over 9.5 °E. Night time scenes are false color composite images using the night microphysical product from EUMETSAT (Eumetsat, 2009). In this product a red, green, blue (RGB) composite where the red channel is the difference between 12.0 and 10.8 μm channels (linear stretch -4 to 2 K), green is the difference between 10.8 and 3.9 μm channels (linear stretch 0 to 10 K) and blue is the 10.8 μm channel (linear stretch 243 to 293 K). Day time scenes are an RGB of the visible channels (R:VIS 06, G: VIS 0.8, B: IR 1.6).

**3 Results and Discussion**

**3.1 Case Study Description**

Two fog events were observed on consecutive days on 9 and 10 September 2017 (hereafter Case 1 and Case 2, respectively) during the AEROCLO-sA campaign at Henties Bay (Fig. 3). Visibility dropped below 1 km from about 03 to 07 UTC during Case 1. The wind direction was predominantly north-westerly and veering to north-easterly during these days, which is in contrast to the dominant wind direction of south-westerly during the full campaign dates. Wind speed was 2 m s$^{-1}$ or less during the events. Water vapour mixing ratio ranged between 8 and 10 g kg$^{-1}$ during the fog events, but was more than 10 g kg$^{-1}$ during the day. The observed relative humidity remained over 90 % throughout the case study period. A temperature inversion with a strength of about 10 °C was present on both mornings below 1000 m.a.s.l. Satellite images for case 1 indicate an isolated cloud over the study site from 00 UTC and by 03 UTC the fog patch is elongated along the coast line (Fig. 4). No cloud is present over the ocean and this fog patch is not associated with a stratus deck. For Case 2, visibility dropped below 1 km from 04 to 07 UTC. Satellite images indicate that cloud was present over the site from 16 UTC the day before and was associated with stratus over the ocean (Fig. 5). This cloud gradually advected over the land and by 21 UTC the visibility had decreased to 2 km. By

03 UTC, just before fog onset, the stratus deck had increased extent over the ocean. After sunrise, the cloud over the land dissipated while the cloud over the ocean remained through the day and into the following night.

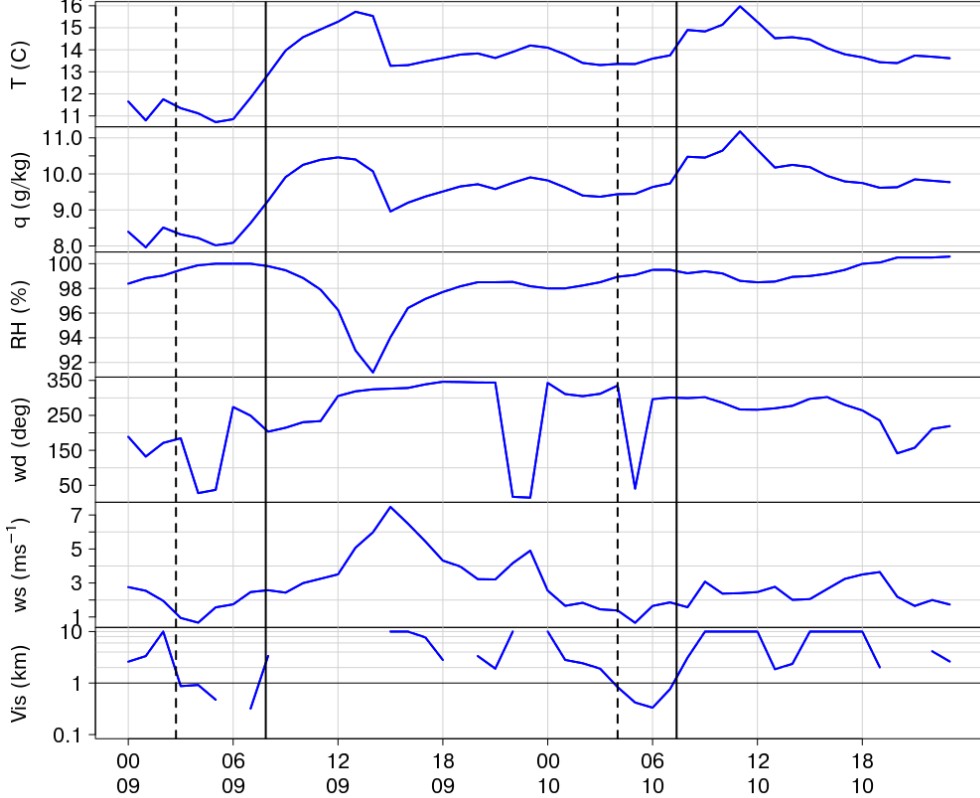

**Figure 3: Meteorological variables from the Cimel Electronique Compact Weather Station at the site during the two fog events on 9 and 10 September 2017. The variables are temperature (T in degrees Celsius), water vapour mixing ratio (q), relative humidity (RH), wind direction (wd), wind speed (ws) and visibility (Vis) from the transmissometer. Vertical black dashed (solid) lines indicate fog start (end) times. X-axis is hours in UTC (Site is UTC+2) and day of month in September.**

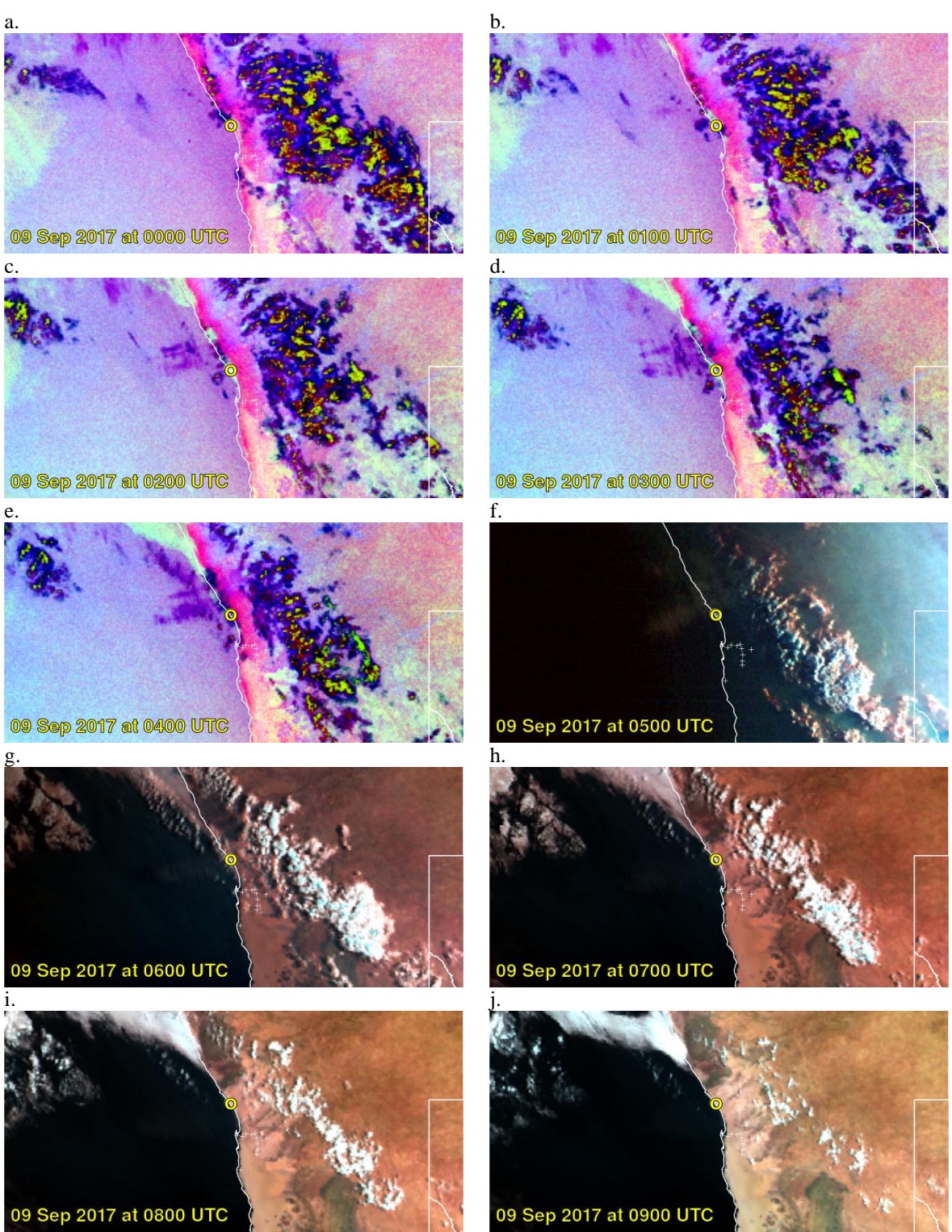

**Figure 4: SEVIRI images of night microphysical RGB product (a-e) and day visible channels (f-j) for the fog event on 2017-09-09. The yellow open circle is the study site. Site is UTC+2. The extent matches the WRF model 3 km domain, which extends 1386 km (east-west) by 720 km (north-south).**

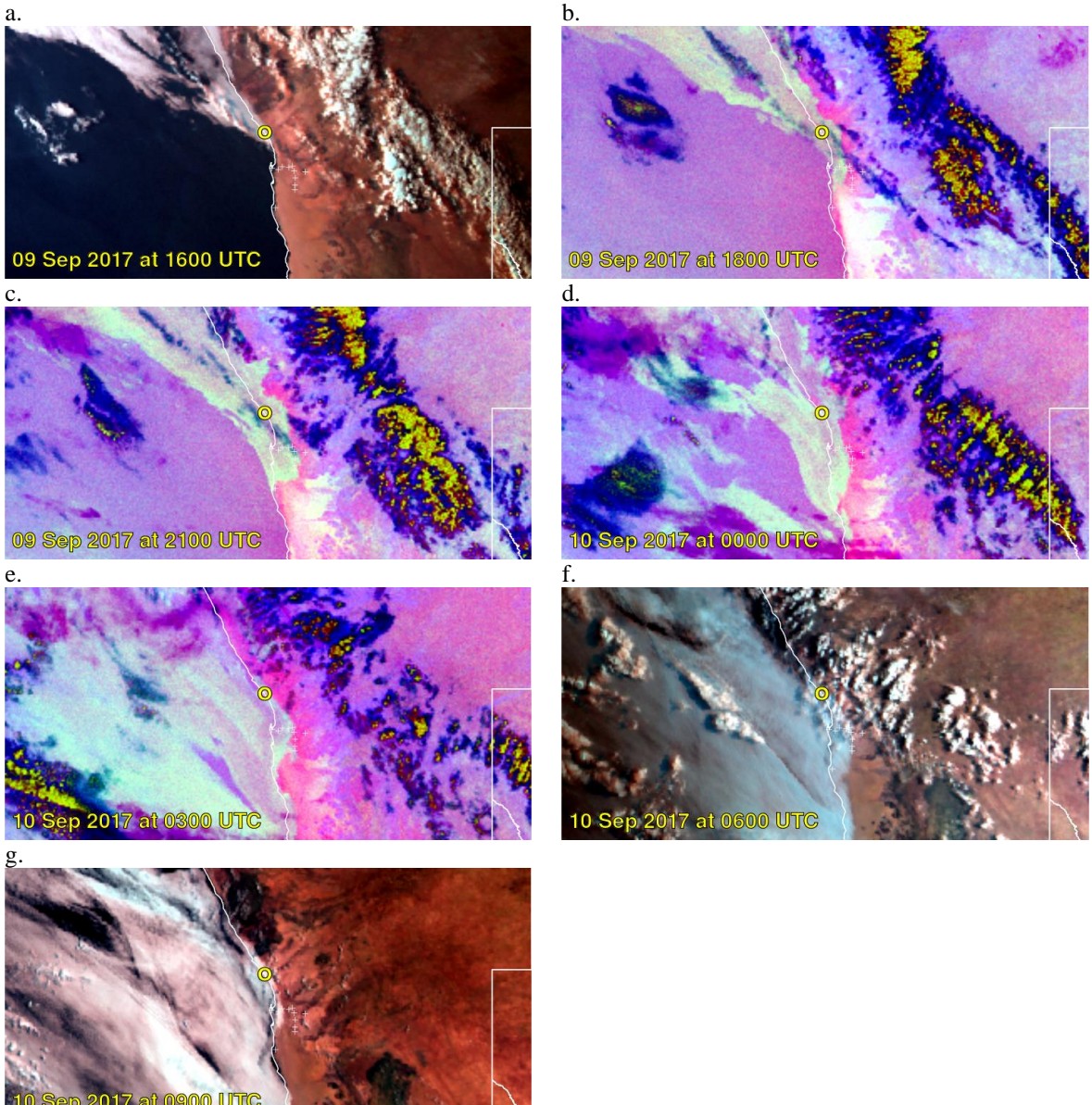

**Figure 5: SEVIRI images of night microphysical RGB product (b-e) and day visible channels (a, f, g) for the fog event on 2017-09-10. The yellow open circle is the study site. Site is UTC+2. The extent matches the WRF model 3 km domain, which extends 1386 km (east-west) by 720 km (north-south).**

## 3.2 Activated CCN

The activated CCN calculated from the observations, as described in section 2.3.2, ranged from 0.058 to 1 (i.e. 5.8 to 100 %), where 1 is the theoretical maximum (Fig. 6). The median (mean) was 0.55 (0.56) and the standard deviation was 0.24. When the fog was most dense, represented by the highest $N_t$ values between 06h30 and 07h15, the activated CCN reached over 80 %. Values of CCN activation of 0.8 at 0.1% super saturation are possible and have been observed (Che et al., 2016). When considering the CCN activation look up table from WRF (Fig. 2), an updraft speed of 0.1 m s$^{-1}$ corresponds to CCN activation just below 0.3. Therefore, assigning a minimum updraft speed of 0.1 m s$^{-1}$ can be a reasonable assumption, as it falls within the median of activation at the site 0.56.

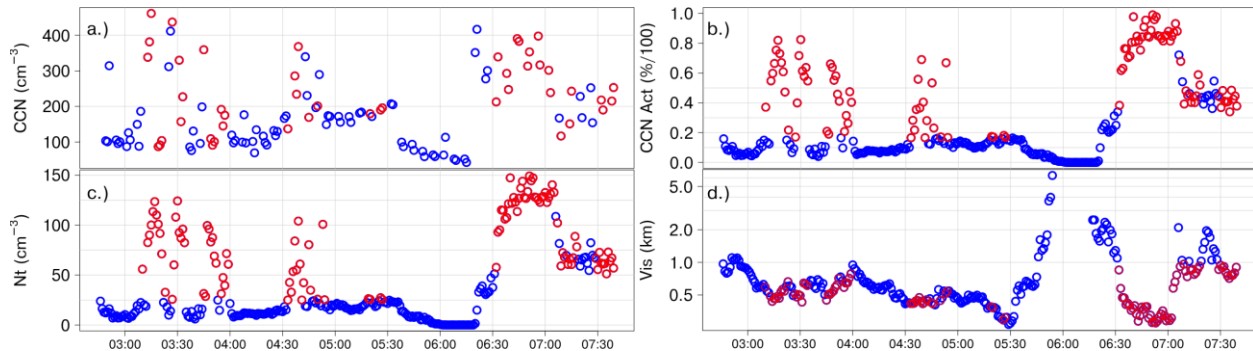

**Figure 6: Timeseries of a.) CCN (cm$^{-3}$), b.) N$_t$ (c m$^{-3}$), c.) Activated CCN (proportion) and d.) Visibility during the fog event on 9 September 2017. Red dots indicate data points used to calculate activated CCN. Blue dots are not used to calculate activated CCN and are shown for context. Times are in UTC (Site is UTC+2).**

### 3.3 Analysis of simulations

### 3.3.1 Evaluation of simulated meteorology

The observed daily temperature range at the study site is narrow and did not exceed5 °C, which is a clear indication of the maritime influence on modulating temperature (Fig. 7a). The modelled diurnal temperature range is larger and more representative of a terrestrial site, with the maximum temperature being 4-5°C warmer and the minimum between 1-5°C colder than observed. The combination of model horizontal resolution and physics may be limited to resolve the exact location of the land-sea interface occurring at the sampling site, which is within 200 m of the sea

shore. Consequently, the cold bias may trigger early saturation, as suggested by the relative humidity time series in Fig. 7e. The water vapour mixing ratio is underestimated by 2 to 4 g kg$^{-1}$ during night time but this is not dry enough to prevent saturation. Simulated wind speed is within 1 m s$^{-1}$ for most of the simulation and more importantly during the observed fog periods (Fig. 7b). However, the maximum bias is 2 m s$^{-1}$ (negative) during the night of 8 September and the model underestimated the daytime wind speed on 9 September by approximately 2 m s$^{-1}$. The observed wind

direction shifts from south-westerly on 8 September to northerly during the case studies. The simulation captures this progression, although the model shifts back to southerly during the day of 9 September (coinciding with the underestimated maximum wind speed) before veering back to northerly.

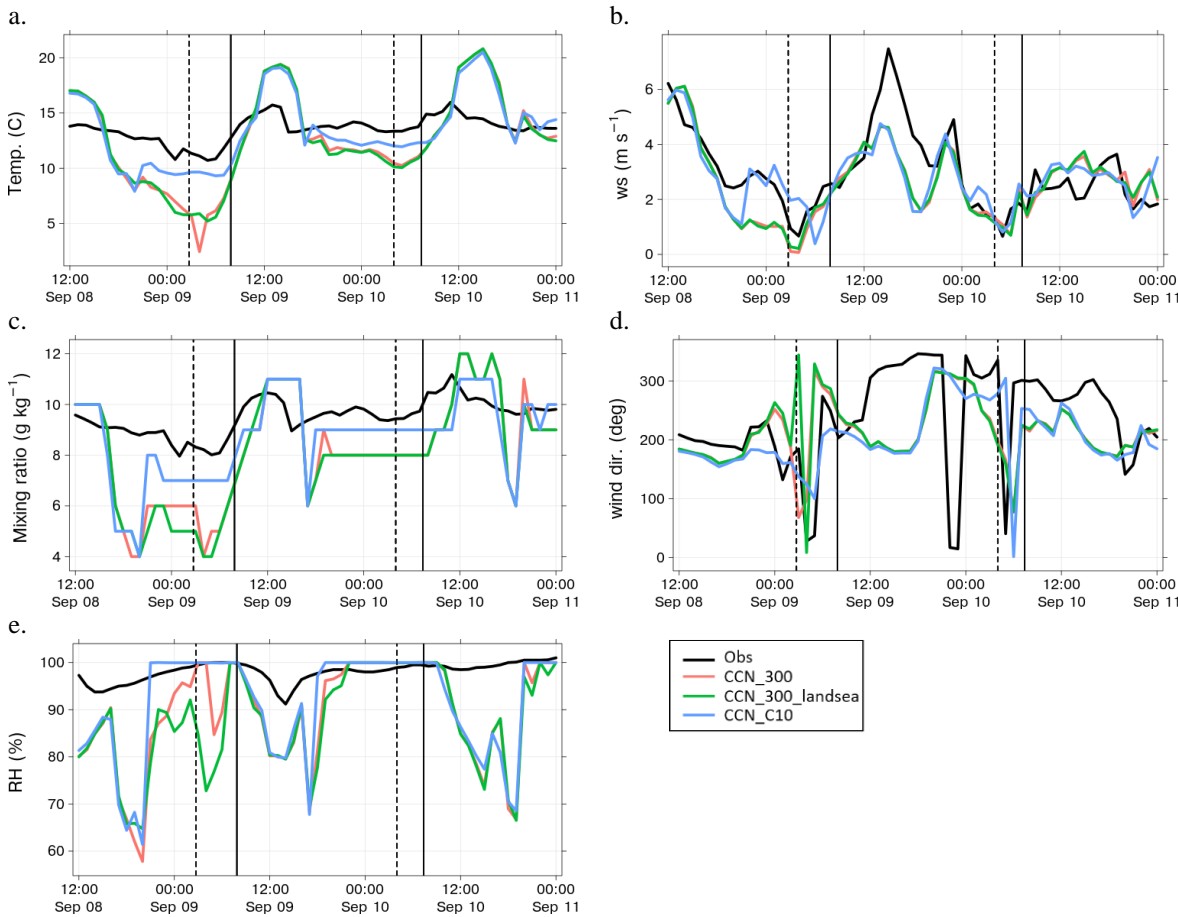

**Figure 7: Timeseries of model and observed meteorological variables at Henties Bay for a.) temperature, b.) wind speed, c.) water vapour mixing ratio, d.) wind direction and e.) relative humidity. Times are in UTC (Site is UTC+2).**

The modelled profiles of temperature and relative humidity displayed the low level inversion and moisture as seen in the observed profiles. For case 1, the model captured the near surface temperature inversion at about the same height 325 (base of inversion is at ~250 m) and strength (~10 °C) as the observed inversion (Fig. 8). Subsequently, the moisture and associated relative humidity was over 80 % near the surface and decreased rapidly from the base of the inversion to 10 % at around 600 m. It is worth noting that the observed relative humidity at the surface was below 100 % and peaked near 100 % at about 250 m, while the model was at 100 % at the corresponding heights. For case 2, the base of the observed temperature inversion was higher at about 800 m and coincided with the peak relative humidity of 330 100 % (Fig. 9). The model inversion height was also higher than case 1 model results but was still below 500 m. The reason for the higher inversion level during case 2 is not clear. This could be attributed to the difference in timing of the sonde (1 hour later), or that the fog event is optically thick and has a major effect on the radiation (as in Price (2011) and Boutle et al., (2018) ). Nevertheless, the model is capturing a low level surface inversion trapping the moisture below this inversion, as in the observations.

a.

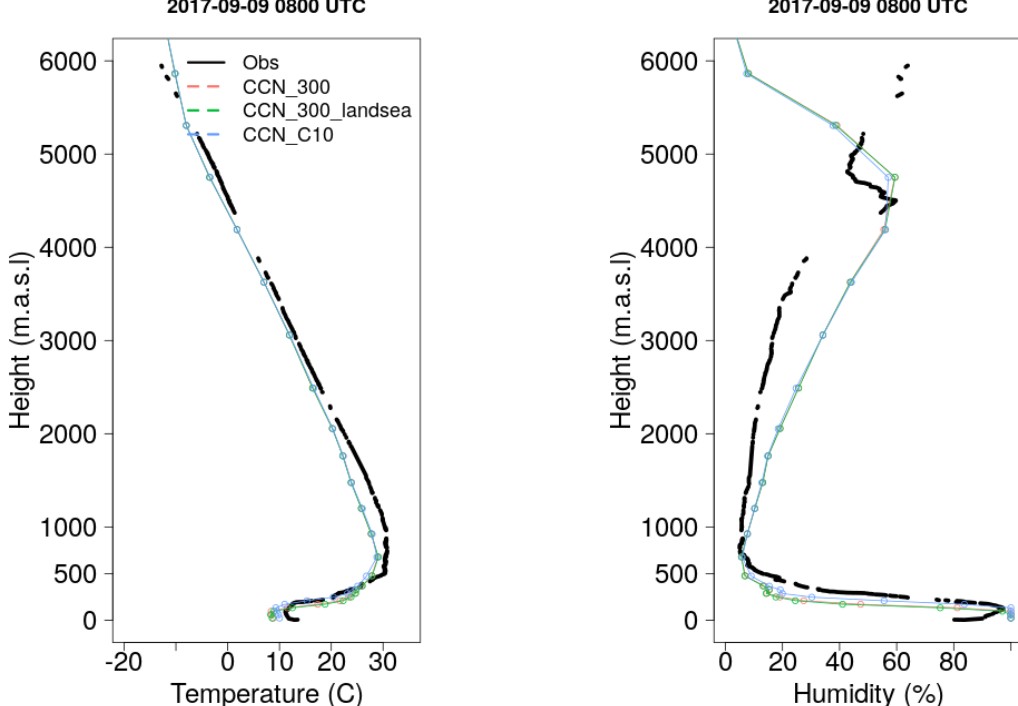

**Figure 8: Radiosonde profiles of a.) temperature and b.) relative humidity from Jakkalsputz on 2017-09-09. The sonde was launched at 07:50 UTC and the model is from 08 UTC. Site is UTC+2.**

a.                                                    b.

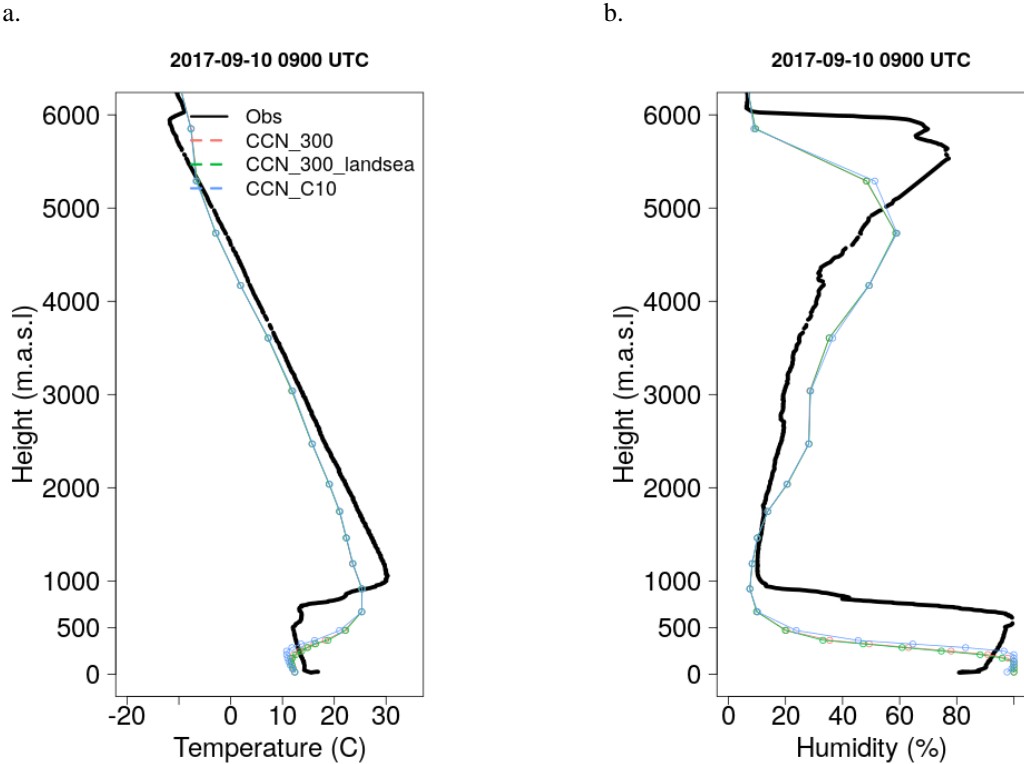

**Figure 9: Radiosonde profiles of a.) temperature and b.) relative humidity from Jakkalsputz on 2017-09-10. The sonde was launched at 09:10 UTC and the model is from 09 UTC.**

### 3.3.2 Spatial distributions of CCN and cloud droplet concentration

The initial CCN concentration for CCN_300 is similar over land and ocean (Fig. 10a). However, over time the concentration over the ocean is relatively higher than over the land, as is evident in the mean concentration in Fig. 10b. This is counter intuitive as observed CCN concentrations are typically lower over ocean than land (Seinfeld and Pandis, 2016). The relative decrease in concentration over the land is most likely due to the treatment of the vertical distribution of CCN in the scheme, where CCN concentrations have a steeper decrease with height when terrain height is above 1000 m. This allows for dilution of the surface CCN concentration during vertical mixing of the atmosphere. Furthermore, the boundary conditions for scenario CCN_300 had relatively lower concentrations of CCN than the ambient CCN in the domain. This explains the lower CCN concentrations over the southern part of the ocean in the domain as clean air was advected from the boundary conditions.

The initial CCN concentration for scenario CCN_300_landsea shows a clear contrast, with lower concentration over the ocean than the land (Fig. 10c). The lower concentration over the ocean counteracts the accumulation of CCN over time, as seen in CCN_300, resulting in a more balanced mean CCN concentration between land and ocean (Fig. 10d). Scenario CCN_C10 has an order of magnitude higher concentrations of CCN (Fig. 10e and f) and does exhibit higher concentrations over land than the ocean. This contrast is maintained though out the simulation as the terrain dependent vertical profile described earlier is not applied to these CCN. Furthermore, as the CCN are a subset of a larger dataset, the boundary conditions include similarly higher concentrations of CCN and clean air is not advected into the domain.

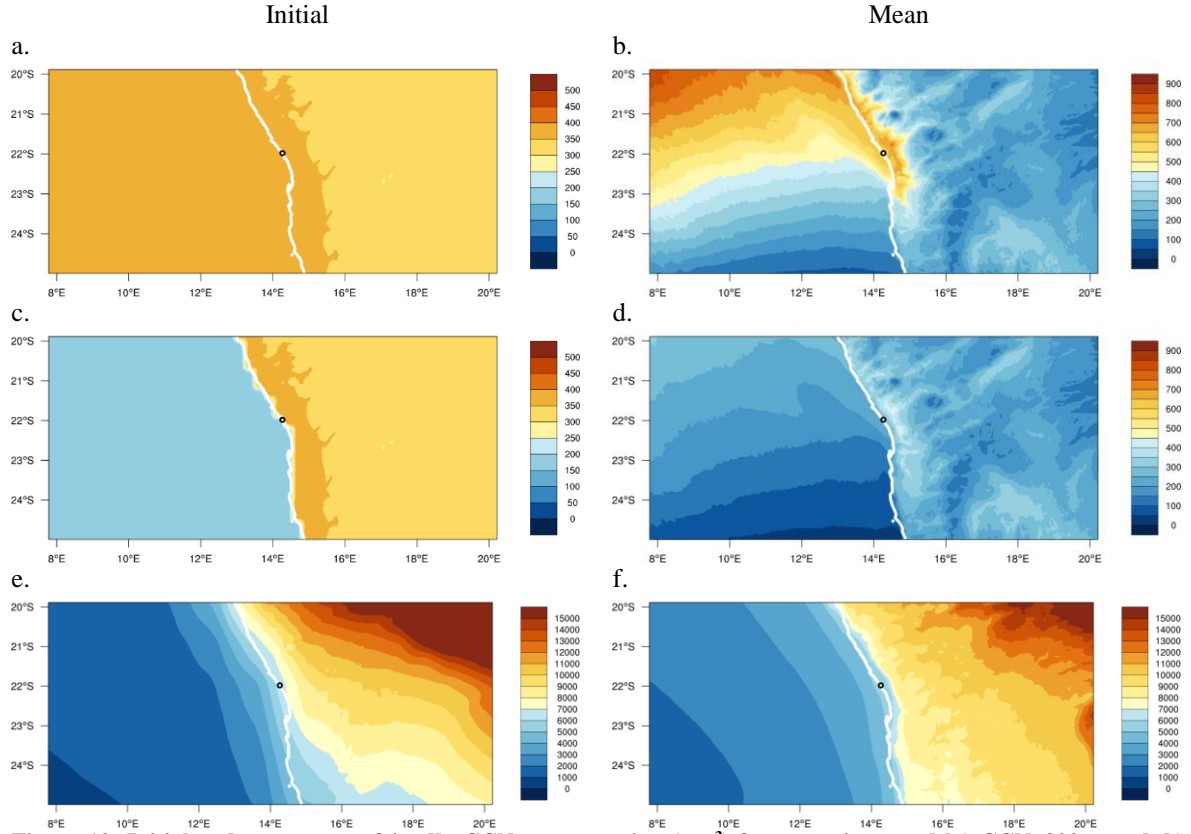

**Figure 10: Initial and mean water friendly CCN concentration (cm$^{-3}$) for scenarios a and b.) CCN_300, c and d.) CCN_300_landsea and e and f.) CCN_C10. Note different scale on c. Black dot is Henties Bay.**

The evolution of CCN concentration for CCN_300 during case 2 is shown in Fig. 11. It clearly shows that by 12 UTC (forecast hour 7) the concentration over the ocean is higher (500-600 cm$^{-3}$) than over the land in the lowest model level. The clean air from the boundary conditions is evident in the south, and continues to propagate through the

domain throughout the forecast. Ahead of this advection, concentration increases as CCN are transported from the south and accumulate. An onshore flow of CCN onto the low lying Namib Desert is also evident. At Henties Bay, the wind direction is south westerly (200 °) from 12 to 18 UTC and veers to north-westerly (300 °) from about 19 UTC to 06UTC. This north-westerly flow allows for the low level transport of the accumulated CCN back along the
coastline.

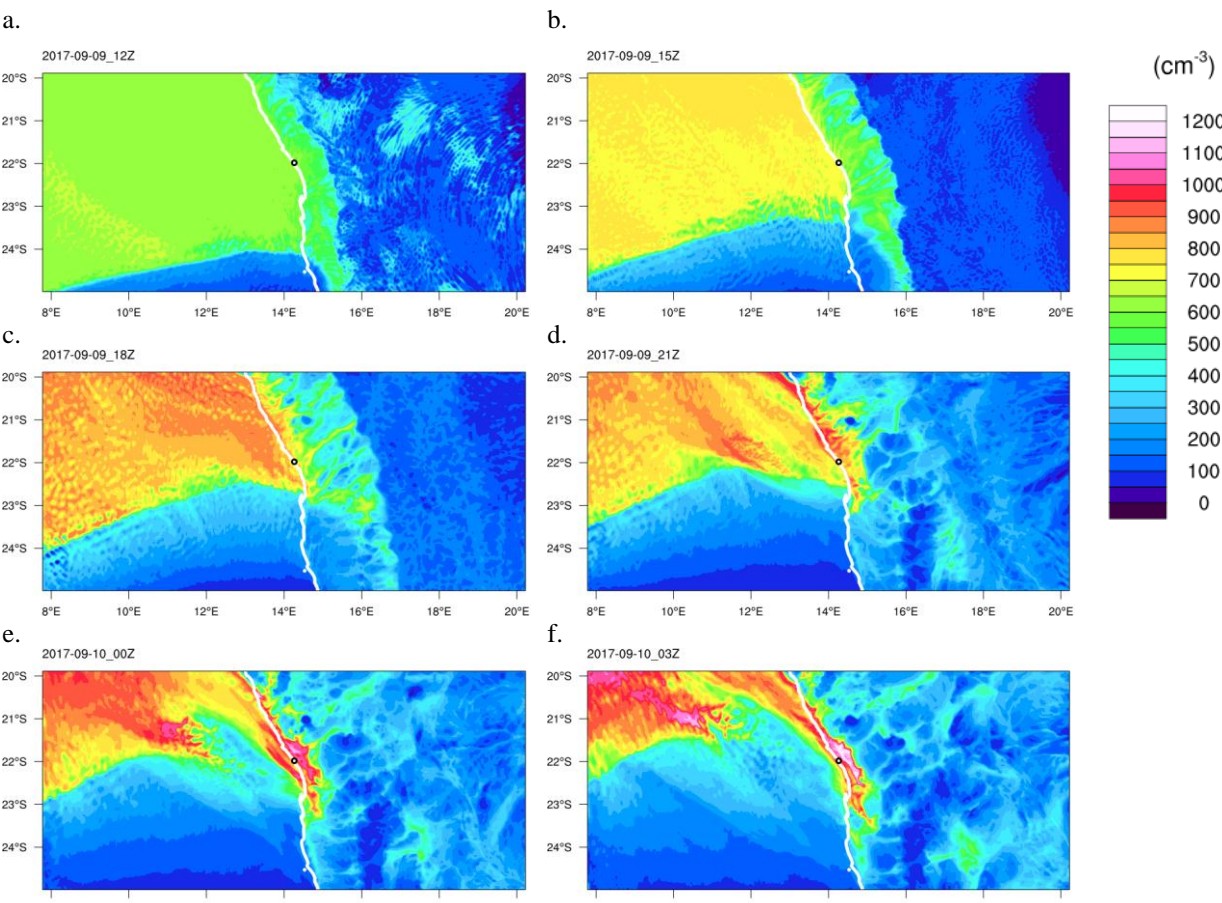

**Figure 11: Evolution of water friendly CCN concentration (cm⁻³) for the event on 2017-09-10 for scenario CCN_300. Black dot is Henties Bay. Time is in UTC (site is UTC+2).**

The fog onset, represented by cloud droplet concentration, is shown in Fig. 12, where droplet concentration reached up to 50 cm⁻³. Over land, fog starts to form along the coast first and then extends further inland, signifying that the simulated fog is due to advection and not to radiation. Radiation fog would be expected to form inland first or simultaneously with coastal fog as the radiative cooling should be stronger inland (e.g. (Weston and Temimi, 2020)).
The fog reached maximum inland extent at 04 UTC (Fig. 13a) before dissipating inland by 07 UTC (Fig. 13d). Cloud droplets are present over the ocean from 21 to 07 UTC. The spatial distribution of CCN and cloud droplet concentration for CCN_300_landsea were similar to CCN_300 although the concentrations were generally lower.

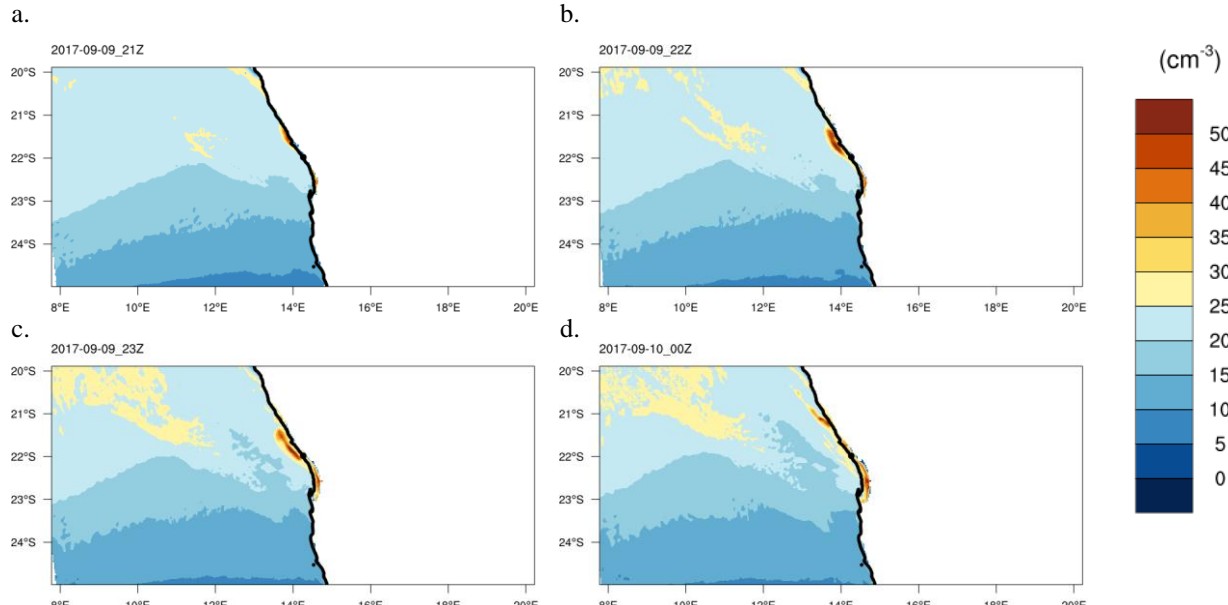

**Figure 12: Cloud droplet concentration (cm⁻³) onset for the event on 2017-09-10 for scenario CCN_300. Black dot is Henties Bay. Time is in UTC (site is UTC+2).**

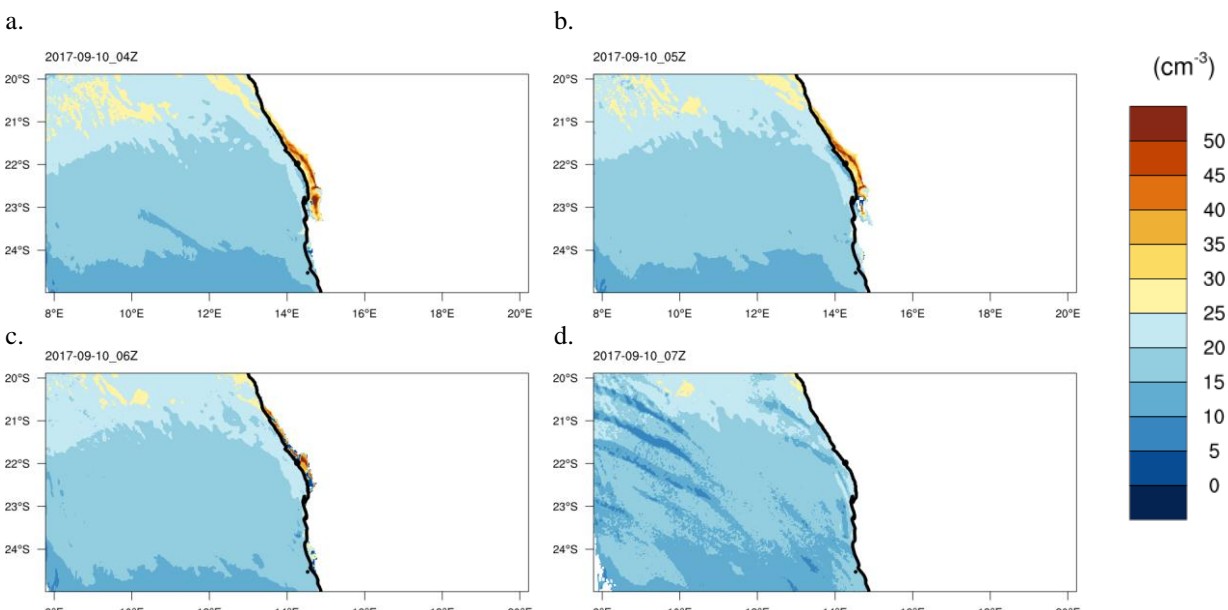

**Figure 13: Cloud droplet concentration (cm⁻³) during maximum extent and highest concentration for the event on 2017-09-10 for scenario CCN_300. Maximum extent was at a.) 04 UTC and maximum concentration at Henties Bay was at d.) 07 UTC. Black dot is Henties Bay.**

The evolution of CCN concentration for CCN_C10 during case 2 is shown in Fig. 14. What is notable is the lack of clean air advection from the boundary conditions, the relative contrast in concentration between land and ocean is maintained throughout and there is generally less variation over time compared to CCN_300. However, the onshore flow and mixing of marine CCN in the Namib Desert is still evident, meaning that the study site is influenced by the marine CCN during these simulations.

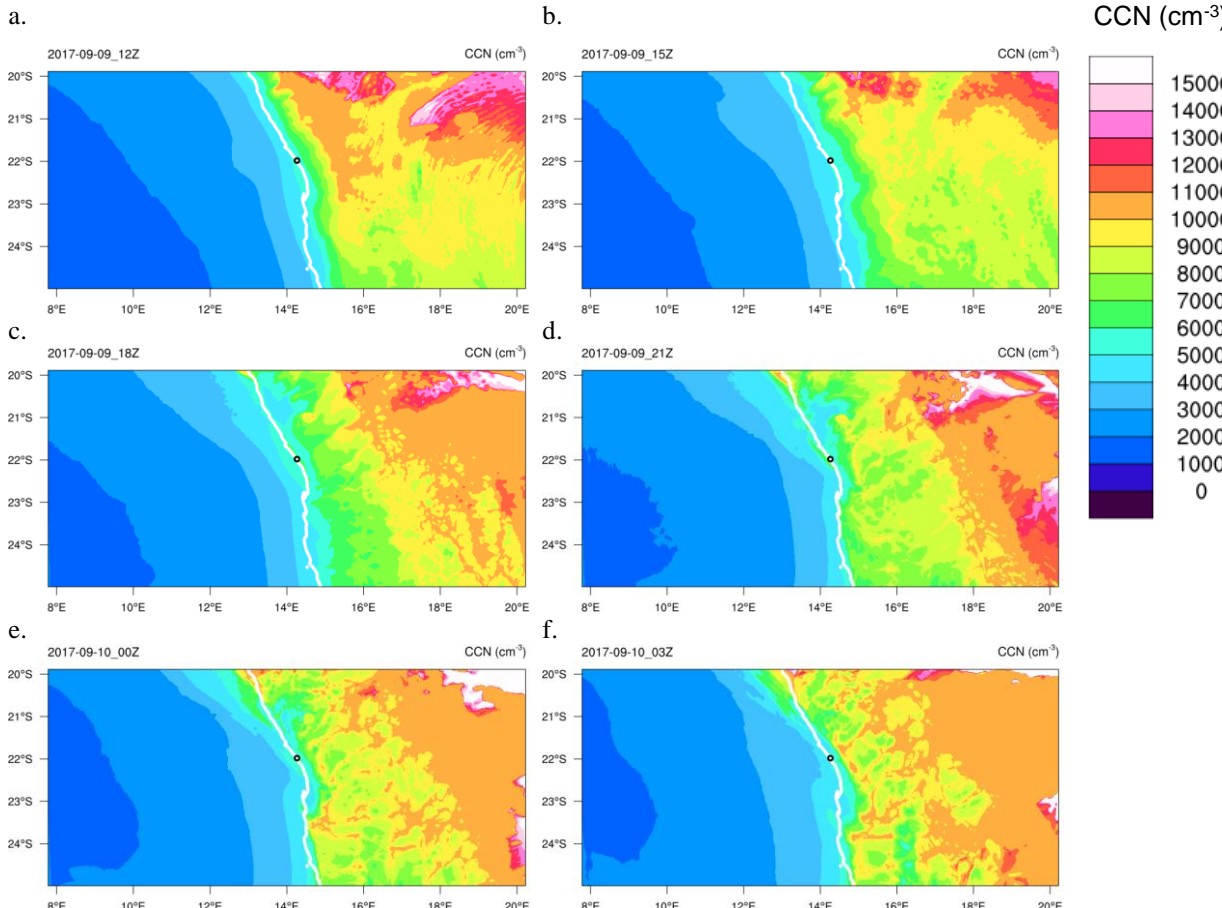

**Figure 14: Evolution of CCN concentration for the event on 2017-09-10 for scenario CCN_C10. Black dot is Henties Bay.**

Fog onset over the land is similar to CCN_300, as this is controlled by the ambient conditions. However, cloud droplet concentration is 2 to 3 times higher in general compared to CCN_300, reaching up to 150 cm$^{-3}$ (Fig. 15). Even though the percentage of droplet activation decreases as CCN concentration increases (Fig. 2a), the initial CCN is significantly high enough to activate more cloud droplets than CCN_300. As in CCN_300, cloud droplets are present over the ocean from 21 to 07 UTC. The maximum fog extent over the land (Fig. 16) is a good match to the satellite data (Fig. 5).

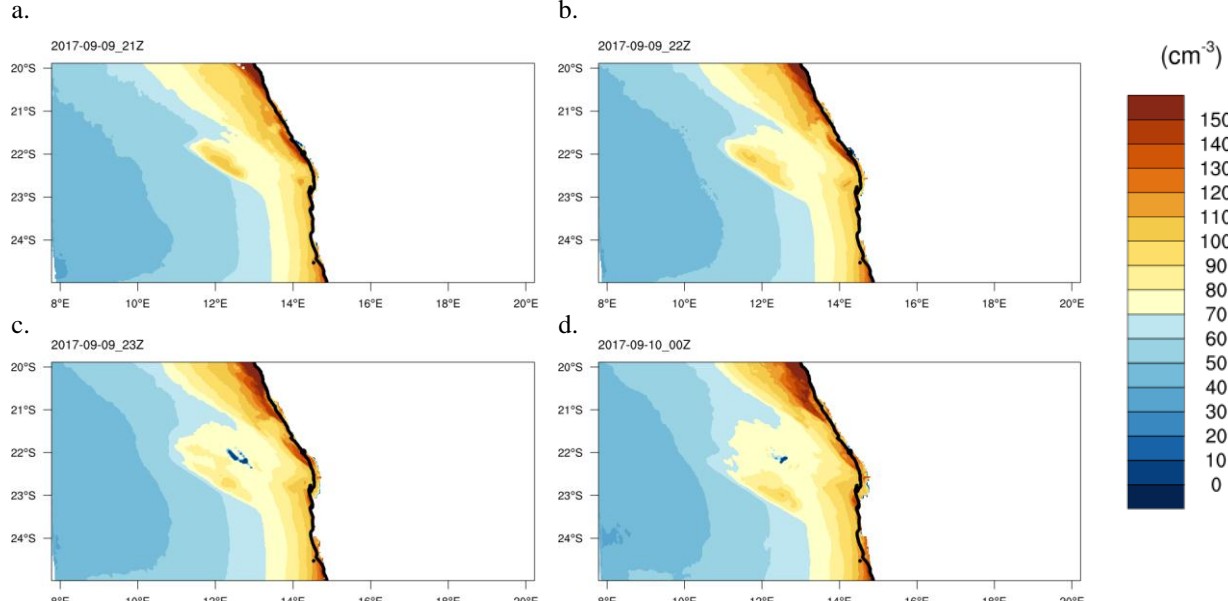

**Figure 15: Cloud droplet concentration (cm⁻³) onset for the event on 2017-09-10 for scenario CCN_C10. Black dot is Henties Bay.**

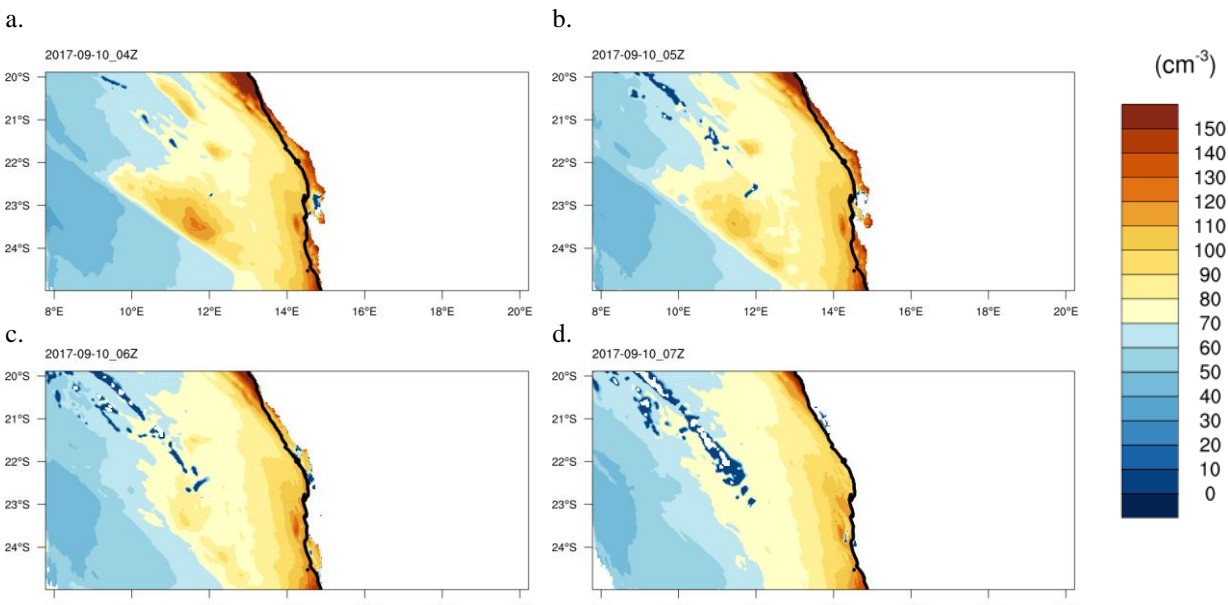

**Figure 16: Cloud droplet concentration (cm⁻³) during maximum extent and highest concentration for the event on 2017-09-10 for scenario CCN_C10. Maximum extent was at a.) 04 UTC and maximum concentration at Henties Bay was at d.) 07 UTC. Black dot is Henties Bay.**

In general, the first three scenarios did well in capturing the fog extent over the land. They show the cloud forming at the coast first before moving inland, suggesting that the dynamics of the formation are captured in the model. Over the ocean cloud droplet activation is overactive, as a cloud deck is present from 21 UTC (Fig. 12 and 15a). At night, the air over the ocean is saturated (not shown) which we assume is from either a cold bias or positive bias in water vapour mixing ratio. Saturation is the first condition that must be met for droplet activation and it is assumed this is

causing a persistent cloud deck over the ocean. The cloud deck was also present in case 1 simulations, while the satellite shows no cloud is present over the ocean. It has been demonstrated now that in all scenarios there is a contrast between the land and ocean in terms of CCN and cloud droplet concentrations. Additionally, the semi-permanent cloud deck in the simulations impinges onto the adjacent land, mainly due to the onshore flow. Similar patterns of

excess frequency of cloud cover (by 30 to 50 %) over the ocean were reported for a regional climate model simulation over Namibia (Haensler et al., 2011). They demonstrate the same high contrast between ocean and land and highlight that the study sites at the coast fall within a transition zone in the model. The study site is located at the interface between these two contrasting conditions and falls within the impingement zone of the simulations. This must be kept in consideration when comparing the model microphysics to the observations from the site, which will be discussed in the following section.

### 3.3.3 Comparison of simulated and observed fog microphysics

The observed microphysics during case 1 showed cloud droplet number concentrations of up to 150 cm$^{-3}$ (mean of 42 cm$^{-3}$), maximum LWC around 0.25 g m$^{-3}$ and MVD up to 27 µm (17a-c). Case 2 demonstrated a higher mean number concentration of 79 cm$^{-3}$, LWC peaked at 0.3 g m$^{-3}$ and MVD was lower up to 21 µm.

During case 1, the cloud droplet number concentration was below 25 cm$^{-3}$ for CCN_300, approximately six times less than the observations. However, the LWC was comparable to the observations at about 0.2 g m$^{-3}$. This means that the LWC is distributed among fewer droplets than the observations and that the DSD will shift towards larger droplet diameters, which is evident in the MVD around 30 µm. The model exhibited larger number concentrations during case 2 which appears to be associated with higher CCN concentrations (17d). This in turn results in marginally increased LWC and an associated decrease in the MVD. As expected, CCN_300_landsea was similar to CCN_300 but with lower cloud droplet concentrations due to lower CCN concentrations. The CCN concentrations were within the observed mean CCN concentration, which could be a good starting point for sensitivity tests on percentage droplet activation for the future. The modelled updraft speed at the site was low, below 0.02 m s$^{-1}$ prior to fog formation and either negative or below 0.01 m s$^{-1}$ during fog (Fig. 17e). This means that the applied updraft speed is the minimum updraft speed from Table 1 when the microphysics is activated.

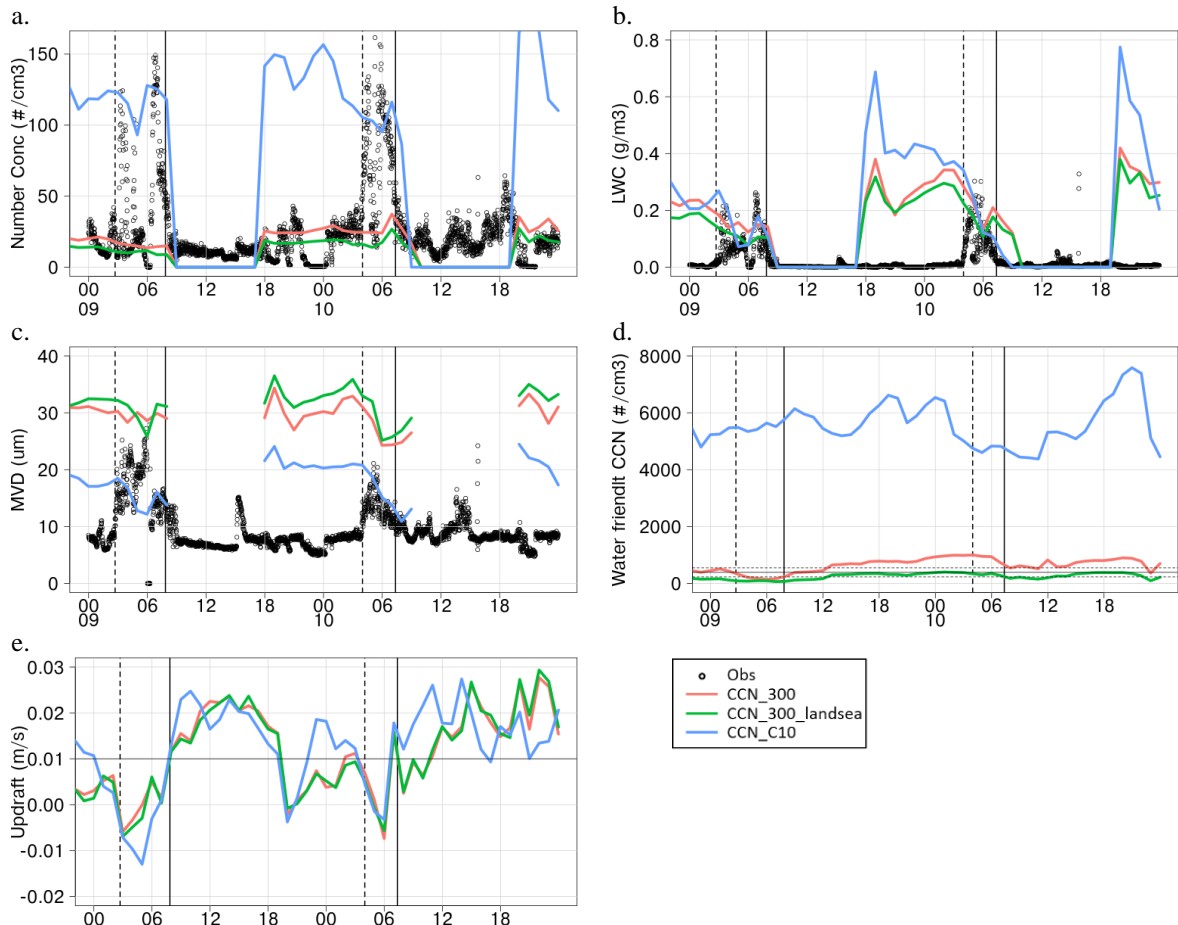

**Figure 17: Observed and modelled a.) $N_t$, b.) LWC and c.) MVD at Henties Bay. Model is hourly and black dots are 1 minute observations. d.) Modelled concentrations of water friendly aerosols. e.) Modelled updraft speed in lowest model level with solid black line at minimum updraft in microphysics scheme. Time is in UTC (site is UTC+2).**

The scenario CCN_C10 has around 10 times higher initial concentrations of CCN and throughout the simulation. This resulted in cloud droplet concentrations around 125 cm$^{-3}$ during case 1 and case 2, which was comparable with the observations. Case 2 was marginally lower than case 1 in this simulation as the CCN were lower. The maximum LWC during case 1 was about the same as the observed at 0.25 g m$^{-3}$ and just over 0.3 g m$^{-3}$ during case 2. As the cloud droplet number concentration and LWC are comparable to the observations, the MVD was much closer with a

maximum around 20 μm for both cases. This scenario demonstrated the best performance in terms of cloud microphysics at the site, which is interesting as the initial CCN values appear to be largely overestimated. The reported mean of CCN at the site ranged from 230-550 cm$^{-3}$ (Formenti et al., 2019), while the modelled CCN was above 4000 cm$^{-3}$ throughout the simulation. As presented in section 0, the observed CCN activation can be between 20 to 50 % during fog events, while the modelled activation is much lower, at around 10 %. To understand this further, we need

to discuss CCN activation in the model scenarios.

CCN activation is most sensitive to CCN concentration and then updraft speed. For each scenario, the CCN concentration is always present and does not vary greatly. Keep in mind that the mean radius and kappa values are constant throughout.

The updraft speed during both cases was negative, meaning that the minimum updraft of 1 cm s$^{-1}$ is applied. If we

consider case 2, we can see that the saturation is met at 18 UTC and droplets are formed, prior to the fog event (Fig. 17a). From this point, droplets persist or are activated whenever saturation is met until the case 2 event. The

potential drawback of using updraft velocity to define drop activation for fog formation has been discussed previously (Boutle et al., 2018; Poku et al., 2019). The argument is that fog often forms under stable conditions when no updraft is present and that activation is instead due to cooling of the air. In addition, the threshold updraft speed is often higher than the 0.01 ms$^{-1}$ used in the T14 scheme, which effectively results in a higher super saturation and excess droplet activation than would be expected for a fog event (Boutle et al., 2018; Poku et al., 2019). Reported activation fractions of CCN in fog are around 20 % but reach up to 40 % (Mazoyer et al., 2019). To some extent, the implementation of a relatively low updraft speed in the look up table mitigates against excess droplets forming in the T14 scheme. Activation at speeds less than 0.03 ms$^{-1}$ are around 10 % or less. Thompson and Eidhammer (2014) acknowledge the use of updraft speed as a potential limitation in the scheme in terms of fog formation. Their proposed work around is to include cooling tendency as proxy for updraft speed and then assigning a speed that will activate the appropriate number of droplets. This may come with a new set of problems in terms of early activation but this remains to be seen. The observed DSD needs to be transformed to a gamma distribution in order to compare it to the model. This is possible as the only inputs to the shape and slope parameters are the number concentration and LWC (Eq. 2 and 3). The mean number concentration and LWC for each case was used and the transformation of the observed (Obs_raw) and gamma distribution (Obs_gamma) can been seen in Fig. 18. While the observations show signs of a trimodal distribution with peaks around 7, 16 and 30 µm, the gamma distribution has one peak at around 11 µm. The three modes in the observed distribution could be representative of the various CCN populations where larger salt ™particles with higher kappa values could represent the largest mode while the smaller sulphates represent the smaller modes. Droplet growth by collision and coalescence can be another explanation for the larger droplets and spectrum widening in a maturing cloud (e.g. (Egli et al., 2015; Mazoyer et al., 2019)). From the model scenarios, CNN_C10 demonstrates the best match with the observations (Fig. 19), with the best performance in case 2. For the scenarios where initial CCN is constant, the distribution shifts to larger droplet sizes.

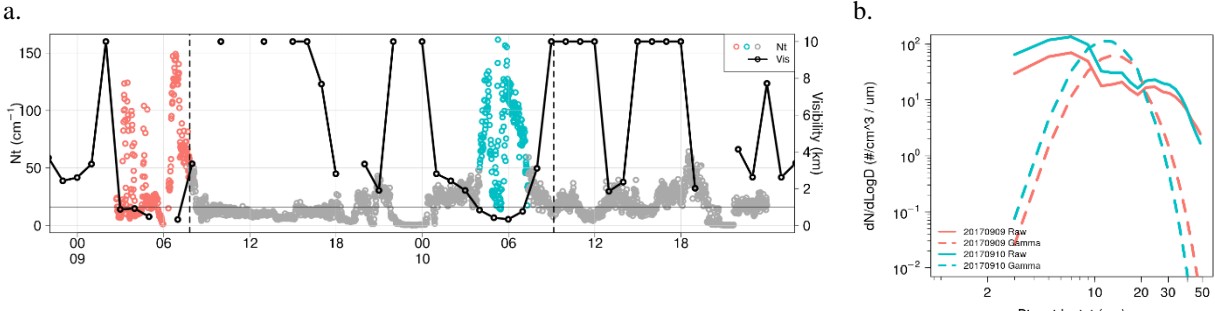

**Figure 18: a.) Timeseries of cloud droplet count (Nt) and visibility. Vertical dashed lines represent sonde release times. Grey horizontal line at 1 km visibility to represent fog conditions. Time is in UTC (site is UTC+2). b.) Droplet size distribution for the two case studies when visibility was 1 km or less. Colours match with the colours in the timeseries. Solid lines are the observed distributions. Dashed lines are the equivalent theoretical gamma distribution applied to the observed data.**

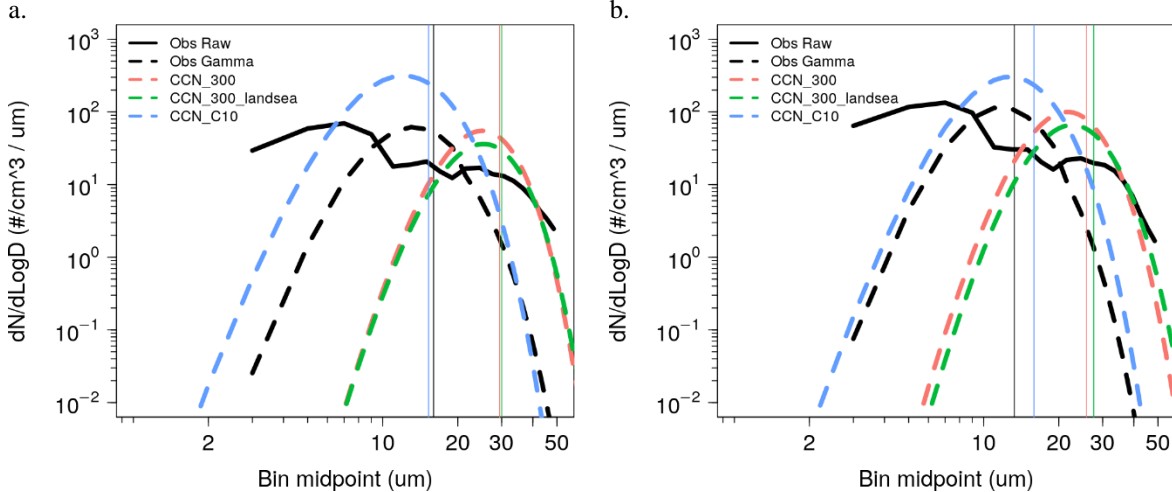

**Figure 19: Observed and modelled DSD for a.) case 1 and b.) case 2. Black solid lines are the observed distributions. Black dashed lines are the equivalent theoretical gamma distribution applied to the observed data. Other dashed lines are model distributions. Vertical solid lines represent the MVD.**

The results of the sensitivity analysis are discussed below. The influence of number concentration on LWC and MVD is further highlighted by CNN_500 scenario. For all times, this scenario has higher cloud droplet concentrations that CCN_300 (Fig. 20a). This results in overall higher LWC and lower MVD (Fig. 20b and c). The inverse is shown with scenario CNN_300_r0.02. In this scenario the radius of the CCN was halved, which results in a lower number of activated droplets according Kohler theory (Fig. 2b). Similar sensitivity analyses have been reported by Poku et al.

(2019) and Stolaki et al. (2015). The most interesting results comes from scenario CCN_300_w0.1, which demonstrated the closest match to the observed number concentration, LWC and MVD. The logic behind this scenario was to increase the number of activated droplets by increasing the minimum updraft speed from 0.01 m s$^{-1}$ to 0.1 m s$^{-1}$, which should allow around 30 % activation. Here we have a scenario where both the CCN and cloud droplet number are in line with the observations. This follows on from the discussion of using temperature tendency as a proxy for

updraft speed, showing that it could work if the nucleation percentage is known, and an appropriate updraft speed assigned. However, in our case, the minimum updraft speed of 0.1 m s$^{-1}$ may relate to an unrealistic cooling rate for this fog event, which we discuss below.

An updraft speed of 0.1 m s$^{-1}$ equates to a cooling rate of 3.51 Khr$^{-1}$ (2.34 Khr$^{-1}$) at the dry (wet) adiabatic lapse rate. Our observed cooling rate at 2 m air temperature is below 1 Khr$^{-1}$ prior to the fog formation (Fig. 3). A cooling rate

of 1 Khr$^{-1}$ equates to an updraft speed between 0.028 to 0.04 m s$^{-1}$ at the dry and wet adiabatic respectively. The modelled updraft speed is below 0.02 ms$^{-1}$ prior to fog formation and therefore in line with the observed cooling rate. Therefore, the use of a minimum updraft speed of 0.1 m s$^{-1}$ does not have a physical basis in our simulation and is used only as part of the sensitivity analysis. It is also clear that if we use a physical basis for the minimum updraft speed that the model will not activate enough cloud droplets and that another physical process must be manifesting

the droplet activation.

In our case, we assume that the observed fog events are due to cloud base lowering (CBL). The satellite images in figures 4 and 5 indicate cloud is present over the site before fog is observed in the surface observations (Fig. 3). The surface observations indicate that fog starts at 02h43 and 04h00 UTC on the 9th and 10th of September respectively (Fig. 3). Cloud is visible from the satellite from 01h00 on 9 September and even 16h00 UTC on 9 September prior to

the fog on 10 September. From the literature, cloud base lowering fog events do not show the same cooling rate at the surface as radiation fog, as the overhead cloud inhibits cooling (e.g. Román-Cascón et al. (2019)). Furthermore, the

influence of the maritime environment at the site will dampen cooling from the desert surface. Thus, our observed

cooling rates are low which is to be expected. The observed droplet number may then be due to the descent of a mature

cloud to the surface, which was initially formed under conditions different to the surface observations. If our fog

droplets are indeed from cloud base lowering, it is possible that our applied updraft speed of 0.1 ms$^{-1}$ may be closer

to the real conditions at the cloud base. While there are no upper air observations at the site to verify this, it could

explain why when applied at the surface the cloud droplet number concentrations in line with the surface observations.

We note that the model does not show this mechanism of CBL. Instead, droplets form in the lowest model level first

and the fog layer thickens over time (as will be shown in section 3.3.4). Thus, the model is missing the initial stratus

cloud formation and subsequent cloud base lowering. As indicated in Figure 7, the model has a cold bias at the surface

and as a result over estimates relative humidity in the lowest model levels (Fig. 9). This highlights the complexity of

the study site which is at the intersection of contrasting land cover and air mass types (ocean and desert) and that the

planetary boundary layer scheme has struggled to simulate this land-sea interface. Adequate modelling of this interface

is perhaps an ambitious task and was not the focus of this study from the outset. As a result the microphysics scheme

has activated at a lower altitude that the observations.

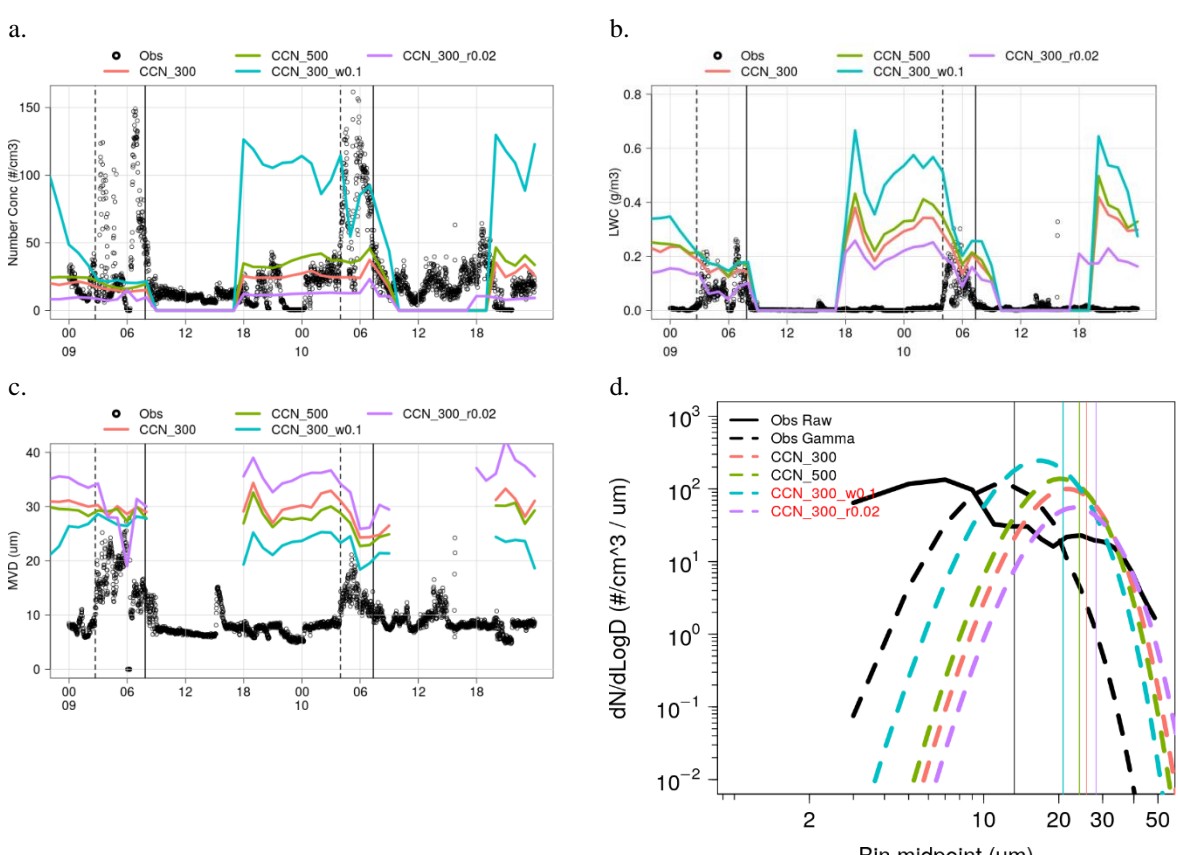

**Figure 20: Observed and modelled DSD for a.) case 1 and b.) case 2. Black solid lines are the observed distributions. Black dashed lines are the equivalent theoretical gamma distribution applied to the observed data. Other dashed lines are model distributions. Vertical solid lines represent the MVD. Time is in UTC (site is UTC+2).**

As has been discussed, cloud droplets occur every night over the ocean in the simulations and this cloud deck impinges

on the land adjacent to the coastline. To avoid this marine/coastal influence, a second point was extracted from the

model about 13 km inland from the study site (labelled "Inland" in Fig. 21). At this point it is clear that saturation and

droplet onset is about 6 hours later than the coastal study site (Fig. 21a). The number concentrations are similar

between the coastal and inland site, while the LWC is lower than the coastal site. This means that the MVD is also

lower and the DSD shifts to the left, resulting in a closer match the observed DSD (Fig. 21). As before, scenario CCN_300_w0.1 is the best match to the observations in terms of number concentration, LWC, MVD and DSD.

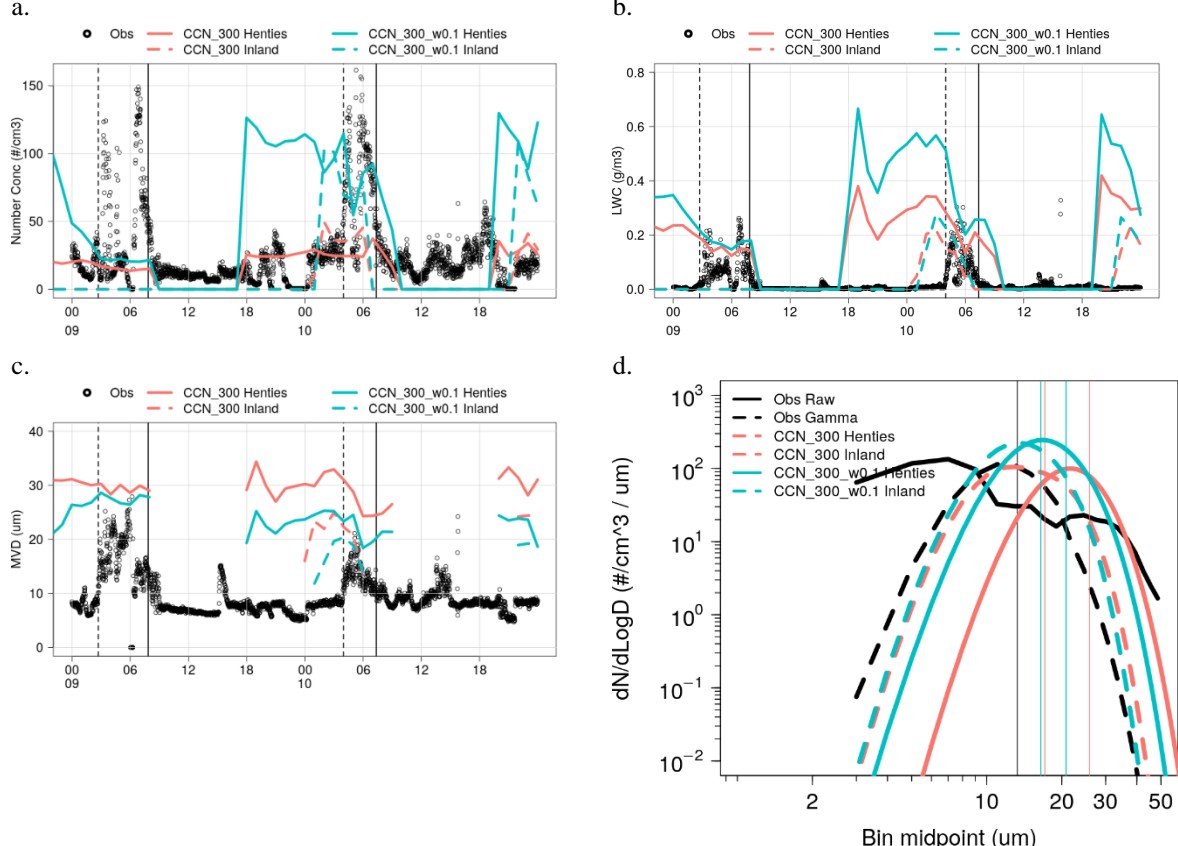

**Figure 21: Observed and modelled DSD for a.) case 1 and b.) case 2. Black solid lines are the observed distributions. Black dashed lines are the equivalent theoretical gamma distribution applied to the observed data. Remaining dashed lines are model distributions. Vertical solid lines represent the MVD. Time is in UTC (site is UTC+2).**

### 3.3.4 WRF vertical evolution

In this section we present the fog top height and the vertical evolution of the fog over time (Fig. 22). It was established previously that the fog onset in the model was associated with advection of marine fog. At the time of fog onset, LWC is only present in the lowest model level and incrementally extends to the layers above. Therefore, the model is not exhibiting any cloud base lowering. Cloud base lowering was deduced to have occurred during case 2, where cloud was observed over the site (Fig. 5) hours before visibility decreased to less than 1 km. Fog top was higher for case 2 than case 1. Fog top was about 100 m for case 1 and 220 m for case 2 for the CNN_300 and CNN_300_landsea scenarios, while CNN_C10 showed fog tops at 220 m and 300 m respectively. Spirig et al. (2019) reported fog top at about 350 m.a.g.l at the Gobabeb site, which is about 56 km inland at an elevation of 406 m. Understandably, the model fog top is lower than this based on the inversion level presented earlier. However, these results demonstrate the benefit of increasing the model vertical resolution near the ground, as it resolves the fog top height.

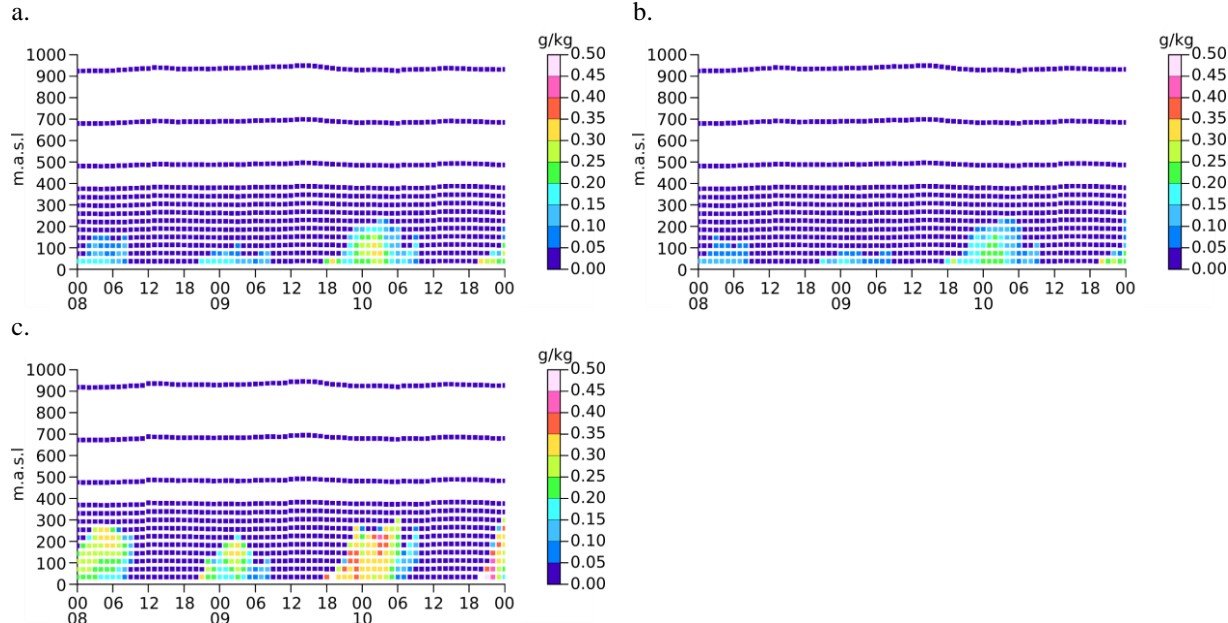

**Figure 22: Profiles of LWC over time at Henties Bay for a.) CCN_300, b.) CCN_300_landsea and c.) CCN_C10. Time is in UTC (site is UTC+2).**

## 4 Conclusions

This study compared WRF model simulations using the Thompson aerosol aware microphysics scheme to observations for two fog cases that occurred during the AEROCLO-sA field campaign at Henties Bay, Namibia. This scheme was designed with cloud microphysics in mind, more specifically clouds where an updraft is present, but can also simulate fog that occurs under stable conditions and in the lowest model levels. A sensitivity analysis was conducted through variations to the initial CCN concentration, CCN radius and the minimum updraft speed, important

factors that influence droplet activation in the microphysics scheme of the model. The first model scenario with initial CCN concentration of 300 cm$^{-3}$ (CCN_300) underestimated the cloud droplet number concentration while the LWC was in good agreement with the observations. This resulted in droplet size being larger than the observations. Another scenario used modelled data as CCN initial conditions (CCN_C10) which were an order of magnitude higher than of the first scenario. However, these provided the most realistic values of $N_t$, LWC, MVD and DSD. From this it was

concluded that CCN activation of about 10 % in the simulations is too low, while the observed appears to be higher, with a mean (median) of 55 % (56%) during fog events. To achieve this level of activation in the model, the minimum updraft speed for CCN activation was increased from 0.01 to 0.1 ms$^{-1}$ for the scenario CCN_300_w0.1. This scenario provided $N_t$, LWC, MVD and DSD in the range of the observations with the added benefit of a realistic initial CCN concentration.

A persistent cloud deck over the ocean was present in the model simulations which impinged on the land immediately adjacent to the ocean. This coincided with the location of the study site and suggests that there is a deficiency in the model physics over water that requires further investigation as it is outside the scope of this study. In order to avoid the influence of the persistent impingement, a model grid point was selected about 13 km in land of the study site and compared to the observations. The timing of the case 2 event was more realistic at this point in the model than at the

study site. This result, albeit for a case study, is encouraging as it suggests that if the outstanding issues of the persistent

cloud deck and land-sea interface can be addressed, that this model setup has the ability to produce realistic timing of events.

The authors acknowledge the limitations of this study due to the low number of cases. However, this is normal for field campaigns over an intensive observation period and the limited cases have demonstrated the benefits of the aerosol aware scheme, especially when parameterised with observations. It is hoped that the results are useful to the modelling community and provide some insight to the model sensitivity in simulating fog.

**Data availability**: Datasets from AERCLO-sA of CCN (DOI: 10.6096/AEROCLO.1813), micrometeorology (DOI: 10.6096/AEROCLO.1808) and radiosonde (DOI: 10.6096/AEROCLO.1806) are stored at baobab.sedoo.fr.

**Author contribution:** MW conceived the study, designed and performed the analysis, and wrote the original draft. SJP and PF contributed to the data collection and curation, conceptualisation, writing, reviewing and editing of the original draft. CD, FB, and SB contributed to the data collection and curation and review of original draft. In addition CD contributed to the methodology of CCN data analysis. TB contributed to data collection and curation. PF and SJP designed the original AEROCLO-sA observational concept and co-led the 5-year investigation.

**Competing interests:** PF is guest editor for the ACP Special Issue "New observations and related modelling studies of the aerosol–cloud–climate system in the Southeast Atlantic and southern Africa regions". The remaining authors declare that they have no conflicts of interests.

**Special issue statement.** This article is part of the special issue "New observations and related modelling studies of the aerosol–cloud–climate system in the Southeast Atlantic and southern Africa regions (ACP/AMT inter-journal SI)". It is not associated with a conference.

**Acknowledgements:** The AEROCLO-sA project would have not been successful without the endless efforts of all the research scientists and engineers involved in its preparation, often behind the scenes. Their support and enthusiasm are sincerely appreciated. The support of the SANUMARC, a research center of the University of Namibia in Henties Bay, is been essential and it is warmly appreciated. The strong diplomatic assistance of the French Embassy in Namibia, the administrative support of the Service Partnership and Valorisation of the Regional Delegation of the Paris–Villejuif region of the CNRS, and the cooperation of the Namibian National Commission on Research, Science and Technology (NCRST) were invaluable to make the project happen. The authors thank the AERIS data center for their support during the campaign and for managing the AEROCLO-sA database.

**Financial support:** the AErosols, RadiatiOn and CLOuds in southern Africa (AEROCLO-sA) is project funded by the French National Research Agency under grant agreement n° ANR-15-CE01-0014-01, the French national programs LEFE/INSU and PNTS, the French National Agency for Space Studies (CNES), the European Union's 7th Framework Programme (FP7/2014-2018) under EUFAR2 contract n°312609, and the South African National Research Foundation (NRF) under grant UID 105958.

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
