# Peer review of "Sensitivity analysis of an aerosol aware microphysics scheme in WRF during case studies of fog in Namibia"

_Atmospheric Chemistry and Physics, 2022_

## Author Response (AR1)

Atmos. Chem. Phys. Discuss., referee comment RC1
https://doi.org/10.5194/acp-2022-152-RC1, 2022 ©
Author(s) 2022. This work is distributed under the
Creative Commons Attribution 4.0 License.

[Figure]

**Comment on acp-2022-152**

Anonymous Referee #1

We thank the reviewer for their comments. The comments have been constructive,
informative and will improve the next version of the submission. Please see responses
marked in blue text below.
* * *
Referee comment on "Sensitivity analysis of an aerosol aware microphysics scheme in WRF
during case studies of fog in Namibia" by Michael Weston et al., Atmos. Chem. Phys.
Discuss., https://doi.org/10.5194/acp-2022-152-RC1, 2022
* * *
The manuscript *"Sensitivity analysis of an aerosol aware microphysics scheme in WRF
during case studies of fog in Namibia"* investigates how droplet activation of cloud
condensation nuclei (CCN) affects simulated evolution of fogs. The analysis is done for two
observed fog cases which are simulated using different assumptions for initial CCN
concentrations affect the evolution of fog properties. The paper presents a thorough
investigation of the topic and is within the scope of Atmospheric Chemistry and Physics. The
main issue with the paper is that it is not clear to me, what is the scientific value of the
study and in order to be published, this should be clarified.

Thank you for this comment as the response should help place this study in better context
for the reader. Our response is included below and we will integrate this into the
introduction in the next submission.

**Lines 80-94**
"Microphysics schemes in mesoscale models are designed with cloud formation in mind, and
not necessarily fog formation. Droplet activation is based on a parcel that is lifted and cooled
adiabatically to reach saturation, and is therefore most sensitive to the updraft speed.
However, fog often occurs under stable conditions where updrafts are low in speed or even
negative. Thus, droplet activation is dependent on other processes like non-adiabatic cooling.
This point is highlighted by Boutle *et al* (2018) who observe a cooling rate prior to fog
formation of $1 Khr^{-1}$ which is equivalent to an updraft speed of $0.04\ ms^{-1}$ assuming a wet
adiabatic lapse rate of $6.5 Kkm^{-1}$. In their case, the minimum updraft speed in the microphysics
scheme was $0.1 ms^{-1}$, and thus would over estimate fog drop activation. To address this issue,
Poku et al. (2021) expanded an existing microphysics scheme to allow for non-adiabatic
cooling, which allowed for more realistic cloud droplet number concentration in the simulation.

In this paper, we assess an aerosol aware microphysics scheme that uses a minimum updraft
speed of $0.01\ ms^{-1}$, which equates to a cooling rate of $0.23\ Khr^{-1}$. This minimum updraft speed

should solve some of the over activation issues highlighted by Boutle *et al* (2018) and Poku *et al* (2021). The main aim is to see how this scheme performs, before applying major changes in the code to account for non-adiabatic cooling rates or similar. Furthermore, our study site is located in the tropics which we see this as a benefit to the community at large as most fog modelling studies are focused on mid- to high-latitude sites."

In addition, the following points should be addressed:

Major comments:

1. In many cases, the model setups and their effects on results are explained too ambiguously to be understandable for the reader. I was not able to understand properly the description of the setup for initial CCN concentrations. Section 3.3.2 discusses the vertical profiles of CCN and refers to the WRF user guide for an explanation. However, it is still unclear to me how this initial vertical distribution results in such a large difference in CCN over land and over ocean for Case CCN_300. Is the initial column intergral of CCN concentrations (CCN burden) significantly different depending on the terrain? Is the CCN concentration initialized in the beginning of the spin up or at the beginning of the actual simulation?

**CCN scenarios**

We agree that the description of the CCN vertical profile should be included in the methodology before being discussed in the results. We will introduce these concepts in the methodology in the next submission. We will also expand the description of the various scenarios and the motivation for them.

With regards to the vertical profiles, Fonseca *et al* 2021 (doi: 10.3390/atmos12121687) have documented the equations from the microphysics fortran routine.

$$N(z) = N_1 + N_0 \ EXP\left[-\left(\frac{h(z) - h(1)}{1000}\right) N_3\right] \qquad (1)$$

with

$$N_3 = -\frac{1}{0.8} LOG\left(\frac{N_1}{N_0}\right) \ if \ h(1) \ \leq 1000 \ \mathrm{m} \qquad (2)$$

$$N_3 = -\frac{1}{0.01} LOG\left(\frac{N_1}{N_0}\right) \ if \ h(1) \ \geq 2500 \ \mathrm{m} \qquad (3)$$

$$N_3 = -\frac{1}{0.8 \ COS \ [h(1) \times 0.001 - 1]} LOG\left(\frac{N_1}{N_0}\right) \ if \ 1000 \ \mathrm{m} < h(1) < 2500 \ \mathrm{m} \qquad (4)$$

Fonseca *et al* (2021). *"In the equations above, h(z) is the height of the model level z in meters, with h(1) being the height of the first model level. The constants N1 and N0 are set to 50 × 106 m3 and 300 × 106 m3 for water-friendly aerosols, and 0.5 × 106 m3 and 1.5 × 106 m3 for ice-friendly aerosols, respectively. This definition is based on the premise that aerosols are mostly concentrated in the lowest part of the atmosphere, with a faster decrease with height over the higher terrain, and a profile tailored for the continental United States."*

If we consider different terrain heights (z) and a first model level at z+34 we get the following initial CCN using these equations. Terrain heights of 0 and 1000 m have the same initial CCN, but after 1000 m it decreases. As Namibia has a coastal plain before the escarpment extends beyond 1000 m, the sea and coastal plain has the same initial CCN

near the surface, while he terrain above 1000 m has a lower initial CCN concentration. This explains the two classes for CCN_300 initial CCN in Fig 10a.

[Figure]

If we plot these equations for different terrain heights (z) we get the following curves (Thompson and Eidhammer 2014 also show this curve but for much higher initial CCN concentration). Height intervals (dots) are 100 m. The decrease in CCN concentration between level z and z+1 is larger if the starting terrain height is 1500 m than 0 m. Thus, as vertical mixing takes place over time, we expect the surface concentration over a higher terrain point to decrease more than a lower terrain point. This partly explains why the CCN over land in Fig 10b can decrease more than the CCN over the ocean when compared to the initial CCN.

[Figure]

We hope this explanation is helpful.

**The manuscript was updated in Section 2.2.2 as follows:**

**Lines 167-169:**
"For example, the initial CCN number concentration is set to 300 cm -3 at the lowest model level. This may not be appropriate for a particular study region or event and can be adjusted accordingly"

**Lines 184-198:**

"Scenario 1 used the default initial CCN values of 300 cm $^{-3}$ near the surface and 50 cm $^{-3}$ in the free troposphere (hereafter CCN_300). The scheme applies a vertical profile to CCN concentration allowing concentrations to decrease exponentially from the surface to the free troposphere (Fonseca et al., 2021; WRF Users Page, 2020)). In summary, the scheme assigns highest CCN concentration near the surface and follows an exponential decrease through the boundary layer to the minimum bound (50 cm $^{-3}$ in our case) which is then assigned to the lower free troposphere. The depth of the boundary layer is made to vary for different terrain heights, ranging to about 1000 m at sea-level to less than 100 m where terrain is greater than 2500 m. The thinner boundary layer at increased terrain height would have a steeper drop off in CCN concentration with height and subsequent dilution during day time mixing. In CCN_300 the initial CCN concentration over land and ocean is the same. Thompson and Eidhammer (2014) proposed different initial CCN concentration for land (300 cm $^{-3}$ ) and ocean (100 cm $^{-3}$ ) based on observations, as ocean air is generally cleaner and contains fewer CCN than continental air (Seinfeld & Pandis, 2016). We implemented this proposal for scenario 2 (hereafter CCN_300_landsea). The same treatment is applied to the vertical profile of CCN concentration as CCN_300. Scenario 3 is the default model setting using the 3-D climatology CCN modified from Colarco et al. (2010) (hereafter CCN_C10). As this is a 3-D data set, no idealised vertical profile is applied at initialisation."

**Lines 203-210**

"Initial results from the study site showed that CCN could reach up to 500 cm $^{-3}$ near the surface (Formenti et al., 2019). Thus, the initial CCN number concentration near the surface was increase from 300 cm $^{-3}$ to of 500 cm $^{-3}$ (hereafter CCN_500), while the free troposphere was kept at 50 cm $^{-3}$ . CCN_500 has the same vertical profile treatment as CCN_300. Initial concentrations over land and the ocean were the same, as in CCN_300. In another scenario, the mean radius of the CCN was decreased from 0.04 to 0.02 μm as part of a sensitivity analysis (CCN_300_r0.02 hereafter). All other setting were identical to CCN_300. In the final scenario the minimum updraft speed was increased from 0.01 m s $^{-1}$ to 0.1 m s $^{-1}$ (CCN_300_w0.1). This motivation for this scenario was to push the model to a higher CCN activation and see how this effects the size distribution results."

2. Page 15, Line 310 says that *"The initial CCN concentration for scenario CCN_300_landsea shows a clear contrast, with lower concentration over the ocean than the land (Fig. 10c). The lower concentration over the ocean counteracts the accumulation of CCN over time, as seen in CCN_300, resulting in a more balanced mean CCN concentration between land and ocean (Fig. 10d)."* Is the accumulation of CCN in CCN_300 only because there are more CCN than in CCN_landsea? In what way there is a more balanced mean CCN concentration between land and ocean? The land-sea contrast at the south boundary seems quite high also in CCN_300_landsea.

We assume that accumulation is occurring in both CCN_300 and CCN_300_landsea as the only difference is the initial CCN over the sea. We will try rephrase line 310 to indicate this. However, the accumulated CCN over the ocean in CCN_300_landsea should be about a third of CCN_300. In CCN_300, the mean CCN over the ocean (Fig 10b) is higher than over the land, which we expect is not realistic as ocean air usually has lower CCN concentrations than continental air. While in CCN_300_landsea, the mean CCN concentration over the ocean are lower or similar to the coastal area of Namibia, which we think is more realistic. The boundary conditions were the same for CCN_300 and CCN_300_landsea, which may explain the similar contrast at the southern boundary.

3. The droplet activation parameterization is shown in Figure 2 to be sensitive to CCN concentration and updraft velocity. However, fog formation is also affected by nonadiabatic cooling. Poku et al., (2021) have suggested that instead of using simulated updrafts, it would be better to calculate the change in saturation due to non-adiabatic processes. In the current paper, only the effect of changing the minimum updraft speed was tested. Wouldn't it have been fairly straight forward to for example convert the cooling rates to corresponding updraft speeds to have a more physical representation of fog droplet activation? On Page 21, Line 392 it is said *"Their proposed work around is to include cooling tendency as proxy for updraft speed and then assigning a speed that will activate the appropriate number of droplets. This may come with a new set of problems in terms of early activation but this remains to be seen."* To me this seems a very good approach and if there is a new set of problems, would that point to problems in other physical processes of the model and in itself is not a good justification for not using this approach?

Yes, we agree that the approach of Poku et al 2021 is sound and a good approach moving forward. For our case, we needed to work backwards, and see which updraft would achieve a suitable activation given realistic CCN concentrations. From this we can work backwards to calculate the equivalent cooling rate required to reach this activation.

Both reviewers have made this point that the non-adiabatic cooling rate may be responsible for the cloud droplet activation and we should present the equivalent cooling rate for our applied minimum updraft speed. We thank the reviewers for highlighting this point. Our response is below and will be incorporated into the next version of the submission.

**Lines 457-481**
"An updraft speed of 0.1 ms$^{-1}$ equates to a cooling rate of 3.51 Khr$^{-1}$ (2.34 Khr$^{-1}$) at the dry (wet) adiabatic lapse rate. Our observed cooling rate at 2 m air temperature is below 1 Khr$^{-1}$ prior to the fog formation (Fig. 3). A cooling rate of 1 Khr$^{-1}$ equates to an updraft speed between 0.028 to 0.04 ms$^{-1}$ at the dry and wet adiabat respectively. The modelled updraft speed is below 0.02 ms$^{-1}$ prior to fog formation and therefore in line with the observed cooling rate. Therefore, the use of a minimum updraft speed of 0.1 ms$^{-1}$ does not have a physical basis in our simulation and is used only as part of the sensitivity analysis. It is also clear that if we use a physical basis for the minimum updraft speed that the model will not activate enough cloud droplets and that another physical process must be manifesting the droplet activation.

In our case, we assume that the observed fog events are due to cloud base lowering (CBL). The satellite images in figures 4 and 5 indicate cloud is present over the site before fog is observed in the surface observations (Fig 3). The surface observations indicate that fog starts at 02h43 and 04h00 UTC on the 9$^{th}$ and 10$^{th}$ of September respectively (Fig 3). Cloud is visible from the satellite from 01h00 on 9 September and even 16h00 UTC on 9 September prior to the fog on 10 September. From the literature, cloud base lowering fog events do not show the same cooling rate at the surface as radiation fog, as the overhead cloud inhibits cooling (e.g. Roman-Cascon 2019). Furthermore, the influence of the maritime environment at the site will dampen cooling from the desert surface. Thus, our observed cooling rates are low which is to be expected. The observed droplet number may then be due to the descent of a mature cloud to the surface, which was initially formed under conditions different to the surface observations.

We note that the model does not show this mechanism of CBL. Instead, droplets form in the lowest model level first and the fog layer thickens over time (Fig 22). Thus, the model is missing the initial stratus cloud formation (or advection) and subsequent cloud base lowering. As indicated in Fig 7, the model has a cold bias at the surface and as a result over estimates relative humidity in the lowest model levels (Fig 9). This highlights the complexity of the study

site which is at the intersection of contrasting land cover and air mass types (ocean and desert) and that the planetary boundary layer scheme has struggled to simulate this land-sea interface. Adequate modelling of this interface is perhaps an ambitious task and was not the focus of this study from the outset. As a result the microphysics scheme has activated at a lower altitude that the observations."

**Minor comments:**
Page 6, Figure 2: Is the activation sensitivity the activated fraction of CCN?

Yes. We have rephrased to "Activated fraction is response to …"

Page 11, Line 266: "*Therefore, assigning a minimum updraft speed of 0.1 m s-1 can be a reasonable assumption, as it falls within the median of activation at the site 0.56*" Did you compare the distributions of activated fractions for different minimum updrafts?

No we did not. We imagine the result would yield the same curve as Figure 2b.

Page 15, Line 307: "Furthermore, the boundary conditions for scenario CCN_300 had relatively lower concentrations of CCN." Lower concentrations compared to what? Why are they lower?

Lower than the ambient CCN concentration and therefore has a dilution effect on CCN. Why this is lower is a good question. This must be due to the parent domain, which we will investigate and comment on in the text.

Page 21, Line 391: "*In addition, the threshold updraft speed is often higher than the 0.01 ms-1 used in the T14 scheme, which effectively results in a higher super saturation and excess droplet activation than would be expected for a fog event.*" Please add references to such studies / approaches.

We will move the references from the previous line (Boutle et al., 2018; Poku et al., 2019) to this line.
**Line 425**
"In addition, the threshold updraft speed is often higher than the 0.01 ms -1 used in the T14 scheme, which effectively results in a higher super saturation and excess droplet activation than would be expected for a fog event (Boutle et al., 2018; Poku et al., 2019)"

The motivation for showing Figures 13-16 is not clear to me.

The motivation for these figures is to give some indication of the spatial dynamics which are influencing the site, as most of the results are comparisons with *in-situ* data at the site. These figures are included to help the reader understand the context of the comparison with *in-situ* data.

**Technical comments:**
Fonts in figures are extremely small.

The following figures have been updated. Font size and scale bars have been increased.
* Fig 3 log scale for Visibility

* Fig 6 labels and titles
* Fig 7 axis and label titles
* Fig 11 scale bar
* Fig 12 and 13 scale bar
* Fig 14 scale bar
* Fig 15 scale bar
* Fig 16 scale bar
* Fig 17 Axis titles, date consistency on d and e.
* Fig 22 all fonts

[Figure]

Atmos. Chem. Phys. Discuss., referee comment RC2 https://doi.org/10.5194/acp-2022-152-RC2, 2022 © Author(s) 2022. This work is distributed under the Creative Commons Attribution 4.0 License.

[Figure]

**Comment on acp-2022-152**

Anonymous Referee #2

*We thank the reviewer for their comments. The comments have been constructive, informative and will improve the next version of the submission. Please see responses marked in blue text below.*
* * *
Referee comment on "Sensitivity analysis of an aerosol aware microphysics scheme in WRF during case studies of fog in Namibia" by Michael Weston et al., Atmos. Chem. Phys. Discuss., https://doi.org/10.5194/acp-2022-152-RC2, 2022
* * *
This paper presents an analysis of two well-observed fog cases over Namibia with multiple versions of an aerosol-aware microphysics parametrization in WRF. The result show some interesting issues with the microphysical parametrization and its representation of fog, which are certainly worth reporting, although the manuscript could be clearer in explaining what these issues are and identifying possible further work to address them. I suggest the paper could be suitable for publication with some revision to improve this aspect.

*To address the point on how the microphysics scheme affect fog formation, we will incorporate the following text into the introduction of the paper.*

**Lines 80-94**

*"Microphysics schemes in mesoscale models are designed with cloud formation in mind, and not necessarily fog formation. Droplet activation is based on a parcel that is lifted and cooled adiabatically to reach saturation, and is therefore most sensitive to the updraft speed. However, fog often occurs under stable conditions where updrafts are low in speed or even negative. Thus, droplet activation is dependent on other processes like non-adiabatic cooling. This point is highlighted by Boutle et al (2018) who observe a cooling rate prior to fog formation of $1 Khr^{-1}$ which is equivalent to an updraft speed of $0.04\ ms^{-1}$ assuming a wet adiabatic lapse rate of $6.5 Kkm^{-1}$. In their case, the minimum updraft speed in the microphysics scheme was $0.1 ms^{-1}$, and thus would over estimate fog drop activation. To address this issue, Poku et al. (2021) expanded an existing microphysics scheme to allow for non-adiabatic cooling, which allowed for more realistic cloud droplet number concentration in the simulation.*

*In this paper, we assess an aerosol aware microphysics scheme that uses a minimum updraft speed of $0.01\ ms^{-1}$, which equates to a cooling rate of $0.23\ Khr^{-1}$. This minimum updraft speed should solve some of the over activation issues highlighted by Boutle et al (2018) and Poku et al (2021). The main aim is to see how this scheme performs, before applying major changes in the code to account for non-adiabatic cooling rates or similar. Furthermore, our study site is located in the tropics which we see this as a benefit to the community at large as most fog modelling studies are focused on mid- to high-latitude sites."*

Major points:

The authors appear to view the minimum updraft speed used for CCN activation as a tuning parameter. It is not. Whilst it is fine to adjust this parameter as part of a sensitivity analysis, the authors need to be clearer on the reasons for doing this, i.e. it is highlighting deficiencies elsewhere in the model? If insufficient activation is achieved for the physically based default setup (obtained from parcel model analysis), then one of 2 things must be happening:

1. The model updrafts themselves are underestimated. This point is not mentioned at all in the paper, and should be. Whilst I suspect (as usually happens in fog), the model updrafts are correctly small, it would be worth discussing - especially if you have observations of the near surface vertical velocity variance available.

We have included in the discussion a comment on Fig 17b.

**Lines 401-403:**

"The modelled updraft speed at the site was low, below 0.02 m s -1 prior to fog formation and either negative or below 0.01 m s -1 during fog (Fig. 17e). This means that the applied updraft speed is the minimum updraft speed from Table 1 when the microphysics is activated."

Please see full response to point 2 below. However, to confirm, yes the model updraft speed is correctly small, below 0.02 ms$^{-1}$ prior to fog formation and either negative or below 0.01 ms$^{-1}$ during fog.

[Figure]

2. The updraft is not the process causing the aerosol activation. Whilst this point is mentioned briefly in the paper, it needs to be made clearer, and could be further discussed, e.g. what cooling rate does the change in minimum updraft velocity they try imply, and is this realistic?

Both reviewers have made this point that the non-adiabatic cooling rate may be responsible for the cloud droplet activation and we should present the equivalent cooling rate for our applied minimum updraft speed. We thank the reviewers for highlighting this point. Our response is below and will be incorporated into the next version of the submission.

**Lines 457-481**

"An updraft speed of 0.1 ms$^{-1}$ equates to a cooling rate of 3.51 Khr$^{-1}$ (2.34 Khr$^{-1}$) at the dry (wet) adiabatic lapse rate. Our observed cooling rate at 2 m air temperature is below 1 Khr$^{-1}$ prior to the fog formation (Fig. 3). A cooling rate of 1 Khr$^{-1}$ equates to an updraft speed between 0.028 to 0.04 ms$^{-1}$ at the dry and wet adiabat respectively. The modelled updraft

speed is below 0.02 ms$^{-1}$ prior to fog formation and therefore in line with the observed cooling rate. Therefore, the use of a minimum updraft speed of 0.1 ms$^{-1}$ does not have a physical basis in our simulation and is used only as part of the sensitivity analysis. It is also clear that if we use a physical basis for the minimum updraft speed that the model will not activate enough cloud droplets and that another physical process must be manifesting the droplet activation.

In our case, we assume that the observed fog events are due to cloud base lowering (CBL). The satellite images in figures 4 and 5 indicate cloud is present over the site before fog is observed in the surface observations (Fig 3). The surface observations indicate that fog starts at 02h43 and 04h00 UTC on the 9$^{th}$ and 10$^{th}$ of September respectively (Fig 3). Cloud is visible from the satellite from 01h00 on 9 September and even 16h00 UTC on 9 September prior to the fog on 10 September. From the literature, cloud base lowering fog events do not show the same cooling rate at the surface as radiation fog, as the overhead cloud inhibits cooling (e.g. Roman-Cascon 2019). Furthermore, the influence of the maritime environment at the site will dampen cooling from the desert surface. Thus, our observed cooling rates are low which is to be expected. The observed droplet number may then be due to the descent of a mature cloud to the surface, which was initially formed under conditions different to the surface observations.

We note that the model does not show this mechanism of CBL. Instead, droplets form in the lowest model level first and the fog layer thickens over time (Fig 22). Thus, the model is missing the initial stratus cloud formation (or advection) and subsequent cloud base lowering. As indicated in Fig 7, the model has a cold bias at the surface and as a result over estimates relative humidity in the lowest model levels (Fig 9). This highlights the complexity of the study site which is at the intersection of contrasting land cover and air mass types (ocean and desert) and that the planetary boundary layer scheme has struggled to simulate this land-sea interface. Adequate modelling of this interface is perhaps an ambitious task and was not the focus of this study from the outset. As a result the microphysics scheme has activated at a lower altitude that the observations."

Some specific (but not exhaustive) examples of text that needs addressing in this regard:

L260-267 - this could be clearer in explaining the motivation, i.e. you're picking the minimum updraft to achieve the observed activation, but this doesn't imply that the updraft is the reason for the activation, i.e. it achieves the right answer but for the wrong reason
We believe that our response to the point above (**Lines 457-481** in manuscript) will address this point.

L450-457 - again, need to be clearer here that the right answer is being achieved for the wrong reasons, and add some discussion on what the right way to achieve the desired result could be.
We believe that our response to the point above (**Lines 457-481** in manuscript) will address this point.

Minor points:

L14 - I'd say "is used" rather than "is parametrized".
**Line 14**: Edited as suggested
L73 - spelling of "Boutle"
**Line 73** Edited as suggested

L112 - is 34m really low enough for the lowest model level in fog? Some more discussion on this might be useful - did you do any sensitivity studies to this? It implies that the fog must be at least 34m deep before it is present in the model, which could have significant effects on its early development. I'd suggest the authors look at https://acp.copernicus.org/articles/22/319/2022/ to see what effect the lowest level height can have on model development, e.g. the FV3 results with a lowest level at 21m are quite poor.

Thank you for this interesting reference.

Yes we agree with your assessment and can list this as a limitation in the simulation. Although, we also note that Spirig *et al* (2019) report fog depth of over 200m at Gobabeb in Namibia, where the stratus cloud intersects with the land.

We have included reference to this latest work:

**Line 128**

"Boutle et al. (2022) recommend having a first vertical level less than 21 m and more than three levels below 150 m. However, our set up is in line with the vertical profiles reported in the literature which show that the moisture is trapped below 500 m (e.g. 130 (Andersen et al., 2019; Formenti et al., 2019; Spirig et al., 2019))"

Fig 3, 6 etc - it's usually helpful to plot visibility on a logarithmic axis, due to its highly nonlinear nature, to better show the low visibility events.

**Fig 3 and 6** have been updated to include log axis for visibility.

L300-315 - I don't really understand here how the CCN is evolved in time in the different experiments, so it would be useful to explain this further. My assumption was that it was a prognostic variable, advected by the flow and processed by the physical parametrizations? But the text seems to suggest that it is somehow diagnosed from the boundary layer depth over land - why? And why is this not applied in the simulations where the CCN is initialised from a climatology - What happens to the CCN when the BL depth adjustment is not used? If it's important enough to discuss, why not process the CCN in the same way in all experiments (with or without this BL depth adjustment) for consistency? I think this is just complicating the results for no good reason, and would be better to be consistent.

We agree that the description of the CCN vertical profile should be included in the methodology before being discussed in the results. We will introduce these concepts in the methodology in the next submission. We will also expand the description of the various scenarios and the motivation for them. Basically, excluding CCN_C10, the treatment of the vertical distribution is the same in all scenarios. CCN_10 does not require any assumption of the vertical distribution as it makes use of a 3-D data set as input.

With regards to the vertical profiles, Fonseca *et al* 2021 (doi: 10.3390/atmos12121687) have documented the equations from the microphysics fortran routine.

$$N(z) = N_1 + N_0 \, EXP\left[-\left(\frac{h(z) - h(1)}{1000}\right) N_3\right] \tag{1}$$

with

$$N_3 = -\frac{1}{0.8} LOG\left(\frac{N_1}{N_0}\right) \; if \; h(1) \leq 1000 \, \mathrm{m} \tag{2}$$

$$N_3 = -\frac{1}{0.01} LOG\left(\frac{N_1}{N_0}\right) \; if \; h(1) \geq 2500 \, \mathrm{m} \tag{3}$$

$$N_3 = -\frac{1}{0.8 \, COS\,[h(1) \times 0.001 - 1]} LOG\left(\frac{N_1}{N_0}\right) \; if \; 1000 \, \mathrm{m} < h(1) < 2500 \, \mathrm{m} \tag{4}$$

Fonseca *et al* (2021). *"In the equations above, h(z) is the height of the model level z in meters, with h(1) being the height of the first model level. The constants N1 and N0 are set to 50 × 106 m3 and 300 × 106 m3 for water-friendly aerosols, and 0.5 × 106 m3 and 1.5 × 106 m3 for ice-friendly aerosols, respectively. This definition is based on the premise that aerosols are mostly concentrated in the lowest part of the atmosphere, with a faster decrease with height over the higher terrain, and a profile tailored for the continental United States."*

If we consider different terrain heights (z) and a first model level at z+34 we get the following initial CCN using these equations. Terrain heights of 0 and 1000 m have the same initial CCN, but after 1000 m it decreases. As Namibia has a coastal plain before the escarpment extends beyond 1000 m, the sea and coastal plain has the same initial CCN near the surface, while he terrain above 1000 m has a lower initial CCN concentration. This explains the two classes for CCN_300 initial CCN in Fig 10a.

[Figure]

If we plot these equations for different terrain heights (z) we get the following curves (Thompson and Eidhammer 2014 also show this curve but for much higher initial CCN concentration). Height intervals (dots) are 100 m. The decrease in CCN concentration between level z and z+1 is larger if the starting terrain height is 1500 m than 0 m. Thus, as vertical mixing takes place over time, we expect the surface concentration over a higher terrain point to decrease more than a lower terrain point. This partly explains why the CCN over land in Fig 10b can decrease more than the CCN over the ocean when compared to the initial CCN.

[Figure]

We hope this explanation is helpful.

**The manuscript was updated in Section 2.2.2 as follows:**
**Lines 167-169:**

"For example, the initial CCN number concentration is set to 300 cm $^{-3}$ at the lowest model level. This may not be appropriate for a particular study region or event and can be adjusted accordingly"

**Lines 184-198:**

"Scenario 1 used the default initial CCN values of 300 cm $^{-3}$ near the surface and 50 cm $^{-3}$ in the free troposphere (hereafter CCN_300). The scheme applies a vertical profile to CCN concentration allowing concentrations to decrease exponentially from the surface to the free troposphere (Fonseca et al., 2021; WRF Users Page, 2020)). In summary, the scheme assigns highest CCN concentration near the surface and follows an exponential decrease through the boundary layer to the minimum bound (50 cm $^{-3}$ in our case) which is then assigned to the lower free troposphere. The depth of the boundary layer is made to vary for different terrain heights, ranging to about 1000 m at sea-level to less than 100 m where terrain is greater than 2500 m. The thinner boundary layer at increased terrain height would have a steeper drop off in CCN concentration with height and subsequent dilution during day time mixing. In CCN_300 the initial CCN concentration over land and ocean is the same. Thompson and Eidhammer (2014) proposed different initial CCN concentration for land (300 cm $^{-3}$ ) and ocean (100 cm $^{-3}$ ) based on observations, as ocean air is generally cleaner and contains fewer CCN than continental air (Seinfeld & Pandis, 2016). We implemented this proposal for scenario 2 (hereafter CCN_300_landsea). The same treatment is applied to the vertical profile of CCN concentration as CCN_300. Scenario 3 is the default model setting using the 3-D climatology CCN modified from Colarco et al. (2010) (hereafter CCN_C10). As this is a 3-D data set, no idealised vertical profile is applied at initialisation."

**Lines 203-210**

"Initial results from the study site showed that CCN could reach up to 500 cm $^{-3}$ near the surface (Formenti et al., 2019). Thus, the initial CCN number concentration near the surface was increase from 300 cm $^{-3}$ to of 500 cm $^{-3}$ (hereafter CCN_500), while the free troposphere was kept at 50 cm $^{-3}$ . CCN_500 has the same vertical profile treatment as CCN_300. Initial concentrations over land and the ocean were the same, as in CCN_300. In another scenario, the mean radius of the CCN was decreased from 0.04 to 0.02 μm as part of a sensitivity analysis (CCN_300_r0.02 hereafter). All other setting were identical to CCN_300. In the final scenario the minimum updraft speed was increased from 0.01 m s $^{-1}$ to 0.1 m s $^{-1}$ (CCN_300_w0.1). This motivation for this scenario was to push the model to a higher CCN activation and see how this effects the size distribution results."

Fig 10 - would be helpful to use the same scale for panels a-d, rather than varying the left and right columns
I think we tried this at first but lost detail and meaning in the plot. We will try as suggested and decide if we should update these scales.

Fig 10 - it would be worth discussing somewhere why there is such a large discrepancy between the simple initialisations and the analysis - are the simple setups just really bad for this area of the world, so the analysis is the correct thing to use, or how much do we trust the analysis for fog initialisation?
I think we are trying to see which setup is closest to the observations. As part of the process we came to realise that this was an ambitious task due to the complexity of the fog formation described in our response under the major comments. We came to realise some challenges like the cloud over the ocean, the land-sea interface, the cloud formation and lowering were not modelled well. These issues can most likely be addressed in the model through sound physical parameterisation, but were beyond the scope of our initial aims. Nevertheless, these results are informative and could help guide future work in the region.

---

## Author Response (AR2)

Atmos. Chem. Phys. Discuss., referee comment RC2 https://doi.org/10.5194/acp-2022-152-RC2, 2022 © Author(s) 2022. This work is distributed under the Creative Commons Attribution 4.0 License.

[Figure]

**Minor Review**

**Comment on acp-2022-152**

Anonymous Referee #2

*We thank the reviewer for their follow up comments. Please see responses marked in blue text below.*
* * *
Referee comment on "Sensitivity analysis of an aerosol aware microphysics scheme in WRF during case studies of fog in Namibia" by Michael Weston et al., Atmos. Chem. Phys. Discuss., https://doi.org/10.5194/acp-2022-152-RC2, 2022
* * *
Many thanks to the authors for their clarifications and changes to the paper, I think this has improved it significantly. I have a few further comments based on the responses and changes, that it would be good to address.

1.
It was not clear to me until after reading the revised paper that the main mechanism of observed fog formation in this area was due to cloud-base lowering, and that the model does not reproduce this, choosing instead to form fog from the lowest level upwards, similar to radiation fog.

I think this point could be worth some further discussion - the arguments presented around the link between cooling and activation (i.e. following Boutle et al 2018, Poku et al 2021) may be representative of how the model is trying to form fog here, but aren't really representative of how reality is trying to form fog here. I

Indeed this may be the reason that a higher minimum updraft improves the model results - reality will have done the activation at a cloud base well above the surface, where updraft speeds will naturally be higher, and then brought these droplets down to surface level.

As the model does not do this, in-situ production of the fog will be much weaker (because of the lack of real updrafts near the surface), and increasing the minimum updraft speed improves matters because it is making the activation more similar to what actually happened at the elevated cloud base in reality.

If the authors agree it would be good to clarify this in the text, as it actually gives some justification for increasing the minimum updraft speed - even though it is wrong in the context of how the model is actually forming fog, it could be representative of how reality formed the fog.

We thank the reviewer for this insight. We have expanded the discussion to include the following:

482-484: "If our fog droplets are indeed from cloud base lowering, it is possible that our applied updraft speed of 0.1 ms-1 may be closer to the real conditions at the cloud base. While there are no upper air observations at the site to verify this, it could explain why when applied at the surface the cloud droplet number concentrations in line with the surface observations."

2.
My only other comment relates to the response to the vertical resolution query.

Firstly, I think a more correct appraisal of the Boutle et al. 2022 results would be a lowest level of <10m (ideally <5m) and 6 or more levels below 150m is necessary for adequate radiation fog simulation.

Secondly, the inference that this is not necessary in this case because the dominant formation mechanism is cloud base lowering would be okay, if the model actually simulated cloud base lowering fog.

The fact that it does not, and indeed does form fog from the surface upwards, means that the near-surface resolution is important and needs to be investigated.

I guess the easiest option may be a simulation with vertical resolution similar to the WRF runs presented in Boutle et al 2022, as that would show what effect this is having on the results?

We thank the reviewer for their comments on the vertical resolution.

Firstly, thank you for correcting the recommendations from Boutle et al 2022. We have corrected this in the text.

Secondly, we agree that an increase in vertical resolution should have an improvement on where clouds form near the earth's surface. A higher resolution near the surface should result in improved simulation of moisture and temperature gradients. Branch et al (2020) (doi: 10.5194/gmd-2020-201) highlighted this for WRF simulations over the United Arab Emirates using 100 vertical levels, with 25 levels below 2000m to better capture convective cases. In addition and as pointed out by the reviewer, Boutle et al (2022) highlight the value of having multiple model levels within a fog bank to better represent the physics at the top and bottom of the simulated fog (e.g. FV3-GFS was an outlier as only 2 model levels were present in the fog bank and the discussion on LWP oscillations in SCMs). We appreciate this point and its value. However, we feel that including an additional simulation within the presented sensitivity analysis is beyond the scope of the paper for the following reasons.

1. Initial simulations in preparation for this paper were rejected due to the lack of vertical resolution in the default WRF vertical levels. Apologies for not including this in the first round response. Additional vertical levels near the surface were included specifically to address the issue of resolving fog top. The decision at the time was that the additional levels and associated vertical resolution was suitable for these cases. From the figure below we can see that multiple model levels are present within the fog layer, in line with recommendations by Boutle et al 2022.

[Figure]

a.

b.

Profiles of liquid water content for a.) Default WRF simulation with default vertical resolution and b.) vertical resolution with 11 levels below 500m.

2. We would like to highlight that the simulations in Boutle et al 2022 are single column models (i.e. not real cases as presented in this paper) and large eddy simulations which are high resolution simulations. The authors have previously increased the number of vertical levels near the surface for real cases over the UAE while keeping the horizontal resolution at 3 or 4 km. The UAE is similar to Namibia in that there is low lying area near sea level and a mountain range above 1000m in the east. The increase in vertical resolution lead to numerical instability in the model over the complex terrain. To avoid this we would most likely need to increase the horizontal and vertical resolution, which becomes a separate sensitivity analysis and falls outside the scope of this study. The number of model levels below the inversion layer where moisture is expected to be trapped are indicated in Figure-9 appear to be a suitable number for these cases.

We have expanded the discussion in the methodology as follows and hope it suitably addresses the comments:

126-137: "A total of 50 vertical levels were used with extra vertical levels added near the surface to allow for 11 model levels below 500 m above ground level (a.g.l). This was decided after initial simulations demonstrated that the default vertical resolution was to coarse near the fog top. The mean height of the lowest 5 levels was 34, 71, 109, 146 and 184 m a.g.l. Boutle et al. (2022) evaluated results from large eddy simulation (LES) and single column models (SCM) for a radiation fog case in the United Kingdom and recommend having a first vertical level less than 10 m and six or more levels below 150 m. An increase in vertical resolution is expected to better simulate strong moisture and temperature gradients in the lower troposphere (e.g. (Branch et al., 2020)). However, Ajjaji et al. (2008) reported that an increase in vertical resolution can have the opposite effect and inhibit cloud formation for fog events over the United Arab Emirates, an arid region similar to Namibia, during a WRF real case (i.e. not SCM) simulation. Furthermore, our set up is in line with

the vertical profiles reported in the literature which show that the moisture is trapped below 500 m (e.g. (Andersen et al., 2019; Formenti et al., 2019; Spirig et al., 2019)). "